# Wheat *Ms2* encodes for an orphan protein that confers male sterility in grass species

Fei Ni[1,2,*], Juan Qi[1,2,*], Qunqun Hao[1,2], Bo Lyu[1,3], Ming-Cheng Luo[4], Yan Wang[1], Fengjuan Chen[1], Shuyun Wang[2], Chaozhong Zhang[1,2], Lynn Epstein[5], Xiangyu Zhao[1,3], Honggang Wang[1,2], Xiansheng Zhang[1,3], Cuixia Chen[1,2], Lanzhen Sun[2] & Daolin Fu[1,2,6]

Male sterility is a valuable trait for plant breeding and hybrid seed production. The dominant male-sterile gene *Ms2* in common wheat has facilitated the release of hundreds of breeding lines and cultivars in China. Here, we describe the map-based cloning of the *Ms2* gene and show that *Ms2* confers male sterility in wheat, barley and *Brachypodium*. MS2 appears as an orphan gene within the Triticinae and expression of *Ms2* in anthers is associated with insertion of a retroelement into the promoter. The cloning of *Ms2* has substantial potential to assemble practical pipelines for recurrent selection and hybrid seed production in wheat.

[1] State Key Laboratory of Crop Biology, Shandong Agricultural University, Taian, Shandong 271018, China. [2] College of Agronomy, Shandong Agricultural University, Taian, Shandong 271018, China. [3] College of Life Sciences, Shandong Agricultural University, Taian, Shandong 271018, China. [4] Department of Plant Sciences, University of California, Davis, California 95616, USA. [5] Department of Plant Pathology, University of California, Davis, California 95616, USA. [6] Department of Plant, Soil and Entomological Sciences, University of Idaho, Moscow, Idaho 83844, USA. * These authors contributed equally to this work. Correspondence and requests for materials should be addressed to D.F. (email: dlfu@sdau.edu.cn).

In the last 48 years, the world's population has doubled to 7.3 billion people[1]. Cereal production similarly increased from 1.2 billion tons in 1969 to 2.8 billion tons in 2014 (FAOSTAT data). The increase in cereal production is basically due to the green revolution[2] and the development of hybrid crops[3]. However, ca. 793 million people still lack sufficient food[4], and the global population is projected to exceed 9.7 billion in 2050 (ref. 1). This rapid growth in population challenges the global food supply in which wheat and rice predominate with wheat providing 20% of humanity's calories.

Plant male sterility has been documented in 617 species or species crosses[5]. Genetically, male sterility is classified as either cytoplasmic male sterility (CMS) controlled by extranuclear genes or genic male sterility (GMS) controlled by nuclear genes[6]. Plant male sterility genes are particularly useful for cereal breeding and hybrid seed production. A three-line hybrid rice that uses the CMS trait was introduced in China after 1974 (ref. 7). However, CMS systems typically confer a cytoplasmic penalty on heterosis. In 1995, a two-line hybrid rice was introduced with an environmentally sensitive GMS (EGMS). This hybrid produced 5–10% more yield than the three-line hybrids[7]. If a male sterility gene were cloned, genetic modification could be used to develop improved methods for improved hybrid production in major crops[8,9].

Although heterosis in wheat results in yield increases of 3.5–15%, hybrid wheat is grown on <0.2% of the global acreage[10]. There are over 70 male-sterile cytoplasm systems in wheat[11], but most are used infrequently[10] because of the yield penalty and a paucity of effective fertility-restorer genes. Nearly all hybrid wheat in Europe (>0.2 mha) uses chemically-induced male sterility[10], which adds cost and furthermore, results in inferior hybrids with either poor germination or reduced seedling vigour[12]. Future adoption of hybrid wheat depends on the availability of a more practical male sterile/restorer system.

In wheat, there are at least five GMS genes and three EGMS genes[13]. The premier Ms2 gene, discovered in 1972 in Taigu county, China, appeared as a spontaneous mutation in common wheat '223' (refs 14,15). Ms2 confers GMS, and all wheat lines with Ms2 are called 'Taigu genic male-sterile wheat' (Taigu). Ms2 confers 100% male sterility regardless of genetic background, phytohormones and environmental conditions[16,17]. Ms2 confers genetically dominant male sterility in hexaploid and tetraploid wheat[18], and hexaploid and octaploid triticale[19–21]. Other than male sterility, Ms2 has no other phenotypes, and Taigu pistils are readily cross-pollinated. Consequently, Ms2 seems suitable for developing hybrids and for recurrent selection in multiple autogamous crops[22]. In the late 1980s, Liu et al.[23] developed a dwarf male-sterile wheat with a tight linkage between the dominant dwarf gene Rht-D1c and Ms2 (RMs2). By 2010, RMs2-based systems had produced 42 wheat cultivars planted on a total of 12.3 mha, with an increased grain yield of 5.6 million tons. Today both Ms2- and RMs2-based recurrent selection systems are widely used in wheat-breeding programs in China during the recurrent selection phase. New wheat varieties are often bred in a ten or more generation process that includes recombination of parental lines (which is greatly assisted by male sterility in an autogamous crop), selection of recombinants during the 'recurrent selection' phase, followed by five or more generations of selfing.

In this study, we report the map-based cloning of wheat male-sterile gene Ms2, and the Ms2-based transcriptome and interaction network. The cloned Ms2 could be used to improve global food security by facilitating breakthroughs in plant breeding and hybrid seed production.

## Results

**Ms2 male-sterile phenotype in Taigu lines.** Taigu lines have small anthers (Fig. 1) and abnormal microsporogenesis (Supplementary Fig. 1). In wild-type LM15, anthers grew continuously from the S1 to S4 stages, appearing green initially and yellow-green at maturity. In LM15$_{RMs2}$, sterile anthers ceased growth at S2 when meiosis occurs, remained the same size through the S3 time period, and decreased in size during the S4 time period. Taigu sterile anthers were initially pale green but faded to pale white or greyish by the end of the S4 time period. At the S4 stage, fertile anthers of wild-type wheat bore germinable pollen; Taigu lines have a 'non-pollen type' male sterility[17] (see trinucleate microspores in Supplementary Fig. 1). Although late-season Taigu flowers may have more normally sized anthers[16], they are aberrant and any pollen that is produced is non-viable (Supplementary Fig. 2).

**The Ms2 gene is mapped to a 0.05-cM interval.** We used synthetic hexaploid wheat (Supplementary Table 1) to enrich for D-genome SNPs. On the basis of the wheat 90k SNP chip, Taigu lines had 5.1% and 14.2% SNPs with SW7 and SW41, respectively (Supplementary Fig. 3), which was sufficient for mapping and cloning Ms2. We mapped Ms2 using four BC$_1$F$_1$ populations, including popA, popD, popE and popF (Supplementary Table 1). Using the Aegilops (Ae.) tauschii map[24], we developed nine PCR markers and mapped Ms2 to the Xsdauw8-Xsdauw36 interval in popA (Fig. 2a, Supplementary Fig. 4 and Supplementary Data 1). Although Ms2 is 4 cM distal to Xsdauw36 in popA, it co-segregates with Xsdauw36 in popE and popF (Supplementary Fig. 4), possibly because the RMs2 locus interferes with chromosome recombination. We screened the Ms2-based populations for recombinants in the Xsdauw2-Xsdauw42 region. Using the physical map of Ae. tauschii (Supplementary Fig. 5), we designed close markers, mapped Ms2 to the 0.6 cM Xsdauw20–Xsdauw32 interval using 3,487 popD plants (Supplementary Fig. 4), and precisely mapped Ms2 to the 0.05 cM Xsdauw27-Xsdauw29 interval using 3,826 popA-2 plants (Fig. 2b; Supplementary Fig. 4). After screening 7,441 gametes in popA and popD, we linked Xsdauw20-Xsdauw27-Ms2-Xsdauw29-Xsdauw32.

We constructed a bacterial artificial chromosome (BAC) library of LM15$_{RMs2}$, which contained 706,176 BAC clones for a 5.1-fold coverage of the wheat genome (Supplementary Table 2). We assembled physical maps of the MS2 region in LM15$_{RMs2}$ (Supplementary Fig. 6) and primarily analysed the Xsdauw24-Xsdauw32 interval (ca. 33 kb). Xsdauw24 is in a pseudogene (PsG, related to AK331827 and AK368680 in GenBank) and Xsdauw32 is near ribosomal protein S25 (RPS25, GenBank CJ502521). Six potential genes (PG1 to PG6; Fig. 2c) were predicted between PsG and RPS25 using the Fgenesh programme (v2.6)[25]. By reverse-transcription PCR (RT-PCR), PG5 mRNA could be detected in immature spikes of Taigu lines CS$_{RMs2}$, LM15$_{RMs2}$ and XY6$_{Ms2}$, but not in male-fertile CS, LM15 and XY6 (Fig. 2c). In contrast, the ribosomal protein S25 (RPS25) mRNA was detected in developing flowers of both Taigu and male-fertile lines; we did not detect expression of the other hypothetical genes (Fig. 2c). Thus, PG5 was the strongest candidate for Ms2. PG5 has two alleles: the P1593-type PG5 (PG5$_{P1593S}$; GenBank KX585234) that is completely linked to male sterility in Taigu lines and the P1076-type PG5 (PG5$_{P1076F}$; GenBank KX585235) from male-fertile LM15. Although the mRNA of PG5$_{P1076F}$ was not detected in LM15, it is expressed somewhat in the S2-stage anthers of LM15$_{RMs2}$, albeit at 23% (mean) ± 8.8% (one standard error of the mean, $n = 3$) of the mRNA of PG5$_{P1593S}$ in LM15$_{RMs2}$ as

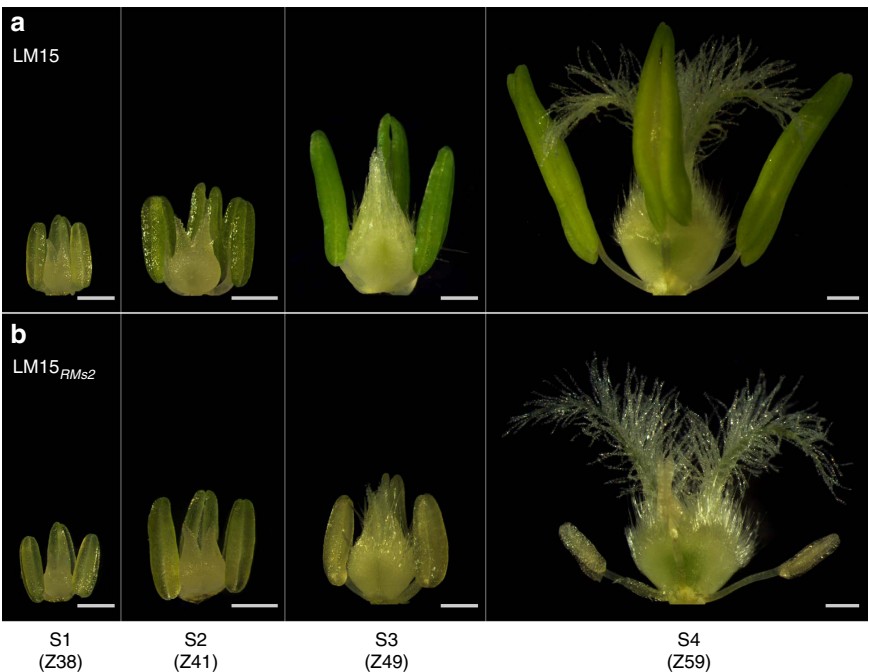

**Figure 1 | Typical anther growth in Taigu lines.** (**a**) Wild-type male-fertile LM15. (**b**) Male-sterile Taigu LM15$_{RMs2}$. LM15$_{RMs2}$ is characterized by diminutive stamens and wild-type pistils. Anther growth was divided into four developmental stages (S1 to S4). Codes in parentheses indicate a comparable Zadok's scale[37]. The experiment was repeated three times. Scale bars, 0.5 mm.

indicated by RNA-seq reads (158 reads of $PG5_{P1593S}$ versus 28 reads of $PG5_{P1076F}$), 5′- and 3′-terminal cDNA clones (10 clones of $PG5_{P1593S}$ versus 4 clones of $PG5_{P1076F}$) and internal cDNA clones (199 clones of $PG5_{P1593S}$ versus 22 clones of $PG5_{P1076F}$). Both $PG5_{P1593S}$ and $PG5_{P1076F}$ have eight exons. $PG5_{P1593S}$ encodes a 293-aa protein (31.5 kDa, pI 5.7) with no annotated domains or protein homologues in plants except wheat.

The $PG5_{P1593S}$ and $PG5_{P1076F}$ alleles of LM15$_{RMs2}$ have seven SNPs in the coding region; the first six SNPs cause six amino acid changes and the seventh encodes a stop codon in $PG5_{P1076F}$ (Supplementary Data 3). We found three structural variations in a 750-bp non-repetitive region upstream of the start codon. In contrast to $PG5_{P1076F}$, $PG5_{P1593S}$ has a long terminal-repeat (LTR) retrotransposon integrated at nucleotide position −314..−310 (Fig. 2d, Table 1 and Supplementary Fig. 7). This LTR retroelement represents a new family of the terminal-repeat retrotransposons in miniature (TRIM), which we named *Taigu_P1593-1* (*Taigu*; GenBank KX585234). *Taigu_P1593-1* is associated with two identical LTRs (direct repeats, 582 bp each), a 630-bp internal domain, and a 5-bp target site duplication (TSD). We postulated that the *Taigu* retroelement acts as a transcription enhancer that activates transcription of $PG5_{P1593S}$ rather than an inactivator of a repressive *cis*-element in the promoter region.

**$PG5_{P1593S}$ confers plant male sterility.** *Ms2* is dominant for male sterility (MS), in keeping with the dominant expression of $PG5_{P1593S}$ in Taigu lines; all Taigu lines are heterozygous for *Ms2* because two Taigu lines cannot cross. We chemically mutagenized LM15$_{RMs2}$ and CS$_{RMs2}$ to determine if any EMS mutants had (1) a male-fertile (MF) phenotype and if so, (2) a variation in their $PG5_{P1593S}$ that would explain the transformation from MS to MF. We generated 2,266 M$_1$ plants and examined 37,471 spikes, of which 254 had MF anthers and

viable pollen (Fig. 3a,b). Among all the EMS-treated LM15$_{RMs2}$ M$_1$ sampled ($n = 1,200$), 2.9% of the primary tillers had a TILLING mutation in $PG5_{P1593S}$, compared to a 40% mutation rate in $PG5_{P1593S}$ in the induced MF tillers ($n = 20$). A $\chi^2$ test negated the independence between $PG5_{P1593S}$ mutations and male fertility in LM15$_{RMs2}$ ($\chi^2 = 97.1$, df $= 1$, $P = 6.5 \times 10^{-23}$), that is, EMS-induced mutation(s) in $PG5_{P1593S}$ are associated with the loss of male sterility. Among all the EMS-treated CS$_{RMs2}$ M$_1$ sampled ($n = 1,066$), we inspected 34,333 spikes and identified 234 MF spikes in 149 M$_1$ plants. On average, each CS$_{RMs2}$ M$_1$ plant produced 32 tillers, but only one to eight tillers were MF, indicating a common phenomenon of chimeracism in the M$_1$ generation (Supplementary Fig. 8). We selected 178 MF M$_1$ tillers and examined the length of $PG5_{P1593S}$ using the HT5 marker (Supplementary Data 4); 29% had either partial or complete deletion in $PG5_{P1593S}$. We then selected 111 of the MF M$_1$ tillers and sequenced the first two exons of $PG5_{P1593S}$; 23% had point mutations that caused either a residue change or a truncation in the PG5$_{P1593S}$ protein (Supplementary Data 5). We then selected 99 MF M$_1$ to represent the range of mutations, selfed them, and examined the M$_2$ (Fig. 3a,b); all were male-fertile, and all had inherited the mutation from the M$_1$. Among 48 MF M$_2$ plants with the $PG5_{P1593S}$ gene, 42 had mutations in the full-length cDNA of $PG5_{P1593S}$, including 34 missense mutations, 3 nonsense mutations and 5 splice site mutations (Supplementary Data 5). Thus, $PG5_{P1593S}$ likely confers male sterility in Taigu lines.

To further corroborate the function of $PG5_{P1593S}$, we performed genetic complementation with plasmid PC976 (Supplementary Table 3), which contained a 10,592-bp genomic fragment (−5578..4078..+936) that included the presumed promoter (5,578 bp), $PG5_{P1593S}$ (4,078 bp) and the terminator region (936 bp) of the gene. PC976 was introduced into wheat, barley and *Brachypodium* (Fig. 3c–e, Supplementary Fig. 9 and Supplementary Data 6). Among 40 putative T$_0$ plants, the expression of $PG5_{P1593S}$ was only detected in three wheat, three

barley, and ten *Brachypodium* plants, all of which were the only male-sterile plants in the T$_0$ generation. Transformants with $PG5_{P1593S}$ remained male-sterile in the T$_1$ generation (Fig. 3c; Supplementary Data 6). We also assembled a genomic $PG5_{P1593S}$ and a green fluorescence protein (GFP) cDNA (PC973; Supplementary Table 3), which was introduced into wheat

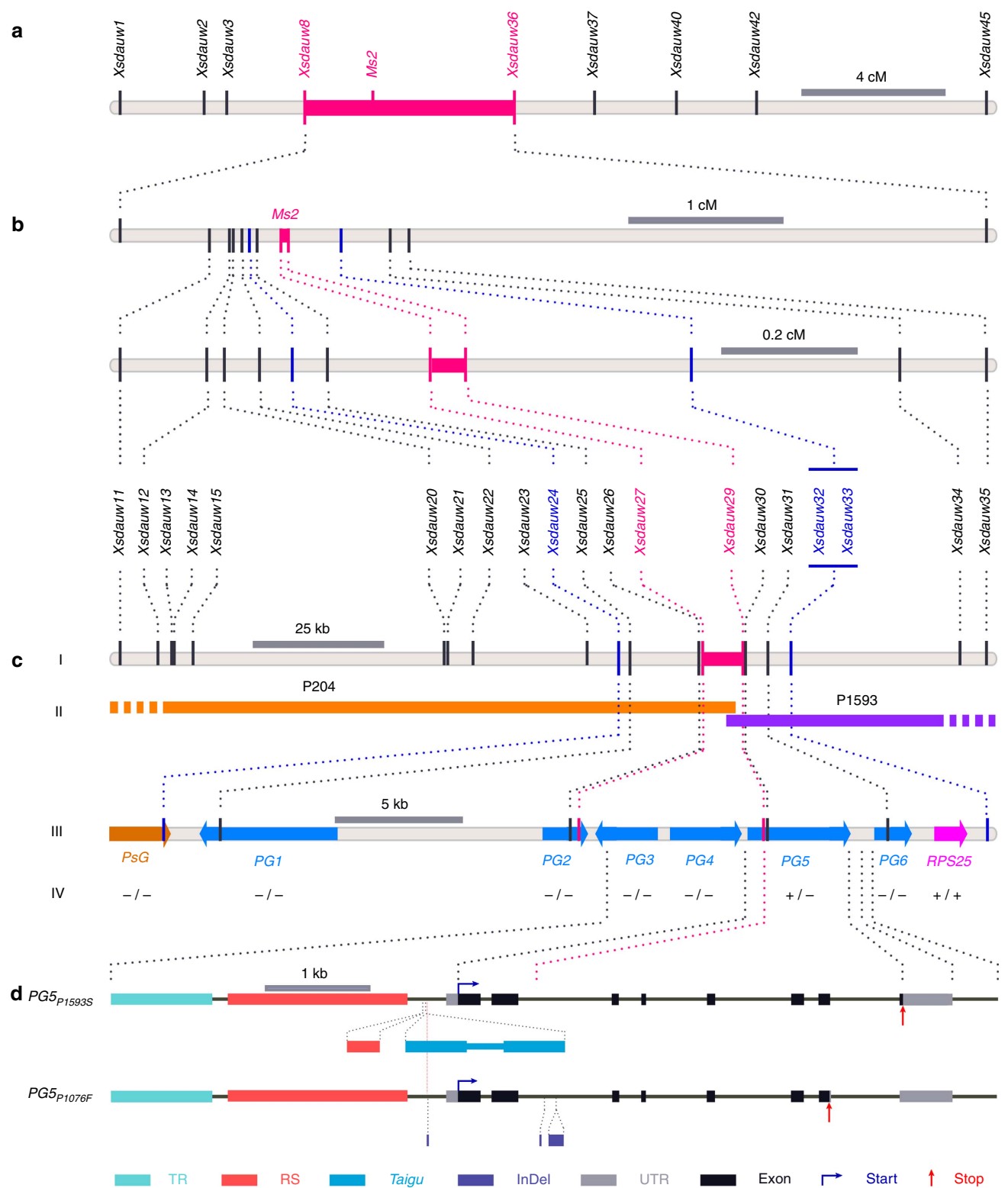

**Figure 2 | Map-based cloning of *Ms2*.** (**a**,**b**) Maps based on popA. (**c**) Physical maps of the *Ms2* region: (I) markers; (II) *Ms2*-associated BAC clones; (III) gene, *pseudogene* (PsG) or potential genes (PG) in the *Xsdauw24-Xsdauw32* interval; (IV) mRNA detected (+) or not (−) in S2-stage anthers of LM15$_{RMs2}$/LM15 (*PG5* additionally tested in *Ms2*-isogenic lines of CS$_{RMs2}$/CS and XY6$_{Ms2}$/XY6; RT-PCR data is provided in Supplementary Fig. 18). (**d**) Genomic structure of $PG5_{P1593S}$ and $PG5_{P1076F}$. Primers for RT-PCR were described in the Supplementary Data 2. InDel, insertion or deletion; RS, repetitive sequence; Start and Stop, respective codons; *Taigu*, a class 1 LTR retrotransposon; TR, tandem repeat; UTR, untranslated-region.

**Table 1 | Haplotype analysis of the *MS2* gene in common wheat and *Ae. tauschii*.**

| Haplotypes | Markers, location and genotypes[*] | | | | Number of lines | | |
|---|---|---|---|---|---|---|---|
| | HT1 −673.. −364 | HT2 −314.. −310 | HT3 902..905 | HT4 3404 | Ae[†] | Ta[‡] | Subtotal |
| CS group | | | | | | | |
| A1 | A | A | A | A | 7 | 60 | 67 |
| A2 | A | C[§] | A | A | 0 | 3 | 3 |
| A3 | A | D[‖] | A | A | 0 | 1 | 1 |
| LM15 group | | | | | | | |
| B1 | B | B | B | B | 31 | 392 | 423 |
| B2 | B | D | B | B | 1 | 0 | 1 |
| B3 | B | — | B | B | 20 | 0 | 20 |
| B4 | B | B | — | B | 0 | 3 | 3 |
| B5 | B | B | B | — | 0 | 4 | 4 |
| Hybrid group | | | | | | | |
| H1 | A | A | A | B | 0 | 1 | 1 |
| H2 | A | E[¶] | B | B | 7 | 0 | 7 |
| H3 | A | D | B | B | 5 | 0 | 5 |
| H4 | B | B | A | A | 1 | 0 | 1 |
| H5 | B | B | B | A | 1 | 1 | 2 |
| H6 | B | — | B | A | 3 | 0 | 3 |
| H7 | B | B | A | B | 1 | 4 | 5 |
| Null group | | | | | | | |
| N1 | — | — | — | A | 20 | 0 | 20 |
| N2 | — | — | — | B | 2 | 0 | 2 |
| N3 | — | — | — | — | 7 | 0 | 7 |
| Sizes[#] | | | | | | | |
| A | 679 bp | 237 bp | 360 bp | 260 bp | NA | NA | NA |
| B | 384 bp | 258 bp | 511 bp | 237 bp | NA | NA | NA |
| | | Number of lines (subtotal and total) | | | 106 | 469 | 575 |

NA, not applicable.
[*]We use two periods (..) to separate the starting and ending nucleotide position (Supplementary Data 3). Regular PCR was performed to genotype the haplotype (HT). Chinese Spring (CS) genotypes are denoted by 'A', and those of LM15 by 'B'. New genotypes are marked by other letters. A dash symbol '—' indicates a negative PCR reaction, possibly caused a target region deletion or low primer efficiency.
[†]*Ae. tauschii* and synthetic hexaploid wheat.
[‡]*Triticum aestivum* (common wheat).
[§]C = 2,037 bp.
[‖]D = 209 bp.
[¶]E = 197 bp.
[#]Major diagnostic bands in base pairs (bp).

(Supplementary Data 6). Among 29 putative $T_0$ plants, the only six male-sterile plants all expressed $PG5_{P1593S}$:*GFP* (Fig. 3f; Supplementary Data 6). This showed that the C-terminal GFP tag did not compromise $PG5_{P1593S}$ protein expression or function. Thus, transgenic studies proved that $PG5_{P1593S}$ confers male sterility in wheat, barley and *Brachypodium*, and that the 10,592-bp fragment encodes for both function and proper spatiotemporal expression of *Ms2*. In conjunction with fine mapping and mutagenesis, we conclude that $PG5_{P1593S}$ controls male sterility in Taigu lines, and that $PG5_{P1593S}$ is the dominant *Ms2* gene.

We tried to overexpress the cDNA of $PG5_{P1593S}$ (PC970), $PG5_{P1076F}$ (PC971) and $PG5_{P1593S}$:*GFP* (PC972) using the maize ubiquitin promoter in wheat, barley and *Brachypodium* (Supplementary Table 3; Supplementary Data 6). In total, we treated 3,474 immature embryos and generated 35 putative transgenic plants, but did not recover any actual transgenics. It is possible that excessive $PG5_{P1593S}$, $PG5_{P1076F}$ and $PG5_{P1593S}$:GFP caused cell death and prevented regeneration of transgenic plants. Despite some nucleotide differences between $PG5_{P1593S}$ and $PG5_{P1076F}$, $PG5_{P1076F}$ appears to be functionally equivalent to $PG5_{P1593S}$, because the overexpression of either $PG5_{P1593S}$ or $PG5_{P1076F}$ apparently prevents generation of transgenic plants.

We postulate that the key difference between the MS and MF alleles is that the MS allele is expressed and the MF allele is not.

**Spatiotemporal expression pattern of *Ms2*.** We did not detect *MS2* mRNA in vegetative and reproductive tissues in male-fertile wheat by RT-PCR (Fig. 4a,b). In Taigu lines, *Ms2* ( = $PG5_{P1593S}$) expression was not detected in early and late in anther development, but was detected in anthers at the S2 stage when meiosis occurs (Fig. 4a). We did not detect *MS2* mRNA in other tissues (Fig. 4b). In Taigu lines, the unexpected transcription of *ms2* ( = $PG5_{P1076F}$) was also limited to the S2-stage anthers. The transgenic *Ms2* in wheat and barley also was detected at the S2 stage in anthers (Fig. 4a,b). Because the $PG5_{1593S}$:*GFP* construct in wheat confers male sterility, the 936-bp terminator region is not required for induction of male sterility.

By *in situ* hybridization, *Ms2* mRNA was detected in Taigu lines in the middle layer, tapetum, pollen mother cells and uninucleate microspores, but not in the endothecium, epidermis and other tissues (Fig. 4c). Ms2:GFP is functionally equivalent to the native Ms2 protein in conferring male sterility

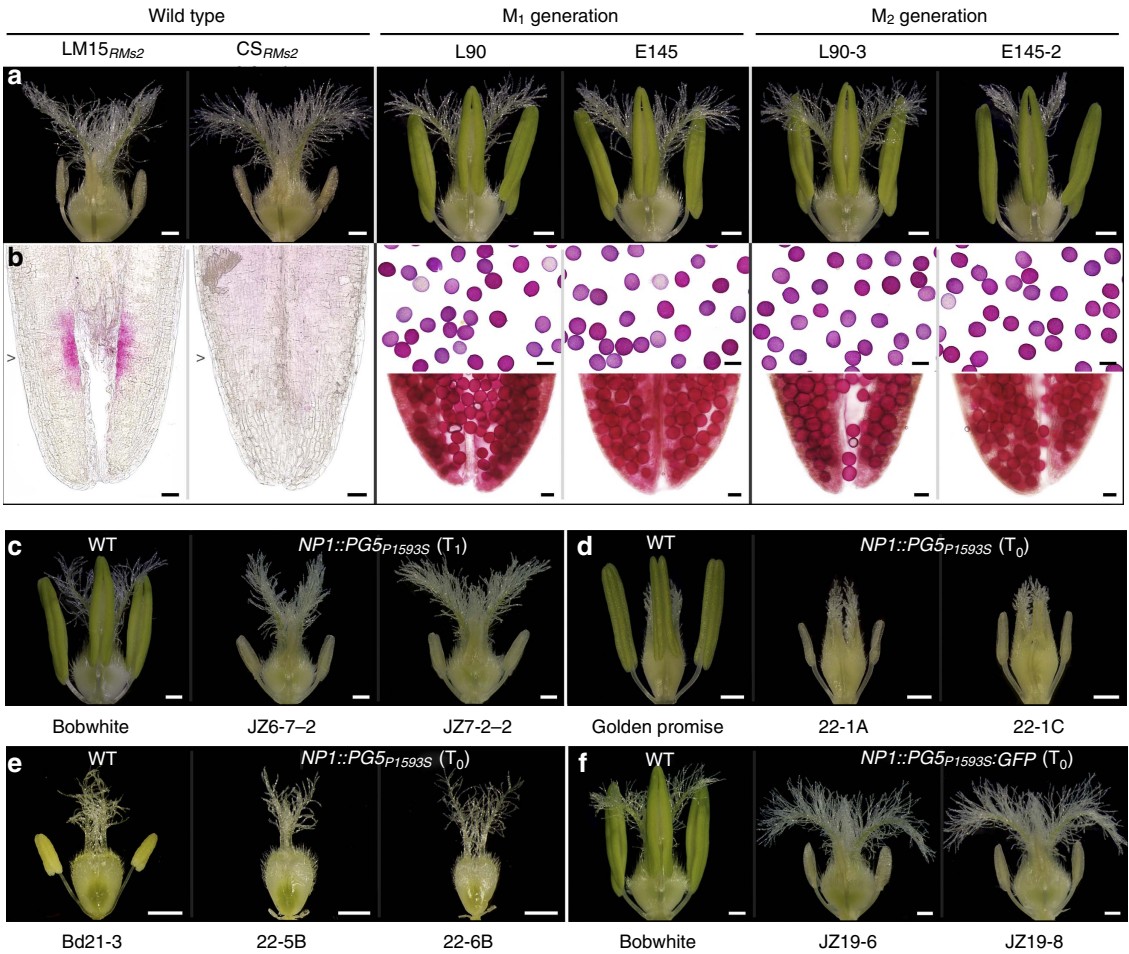

**Figure 3 | $PG5_{P1593S}$ confers male-sterility in Taigu.** (**a**) Mature flowers from $LM15_{RMs2}$ and $CS_{RMs2}$ with diminutive anthers, and $CS_{RMs2}$ mutants with normally-sized anthers. (**b**) Alexander's staining of anthers (lower) and pollen (upper, except when no pollen was produced). Wheat lines are the same as in (**a**). (**c–f**) Flowers in wild-type (WT, on left) and transgenic wheat (**c,f**), barley (**d**) and *Brachypodium* (**e**). Male-sterile transgenic plants at the $T_1$ (**c**) and $T_0$ (**d–f**) generations were positive for the native $PG5_{P1593S}$ gene (PC976; **c–e**) and the $PG5_{P1593S}$:*GFP* gene (PC973; **f**). Scale bars, 500 μm (**a,c–f**) and 50 μm (**b**).

(Fig. 3f; Supplementary Data 6). The distribution of Ms2:GFP and the *Ms2* mRNA also were indistinguishable (Fig. 4d). In mesophyll protoplasts of tobacco, the GFP:Ms2 appeared to have a distribution consistent with cytoplasmic and rough endoplasmic reticulum localization, however, there was no detectable green fluorescence in cells with the Ms2:GFP construct (Supplementary Fig. 10). The reason for this currently remains unclear.

***MS2* appears within the Triticeae.** *MS2* homologues only are present in some Triticeae species, including *Ae. tauschii*, *Triticum urartu* and *T. aestivum* (Supplementary Fig. 11; Supplementary Table 4). The lack of close homologues in *Hordeum vulgare* suggested that *MS2* originated between 2.6 and 8.9 Myr ago[26]. Afterwards, *MS2* apparently duplicated, resulting in at least two paralogues in each chromosome of the homoeologous group 4 in common wheat and the ancestral species of *Ae. tauschii* and *T. urartu* (Supplementary Table 4). Some regions only share similarity with one or a few pseudo-exons, which have additional truncations, inversions and duplications. Interestingly, five pseudo-exons in WGSC_4898666 match one cDNA entry (GenBank AK331503), and two pseudo-exons in WGSC_3034403 match another cDNA entry (GenBank BE500370). We propose that the *MS2*-like regions may have undergone dynamic exon shuffling.

We compared the *Xsdauw7-Xsdauw35* interval in the D-genome in eleven accessions, including seven common wheat, three synthetic wheat and *Ae. tauschii*. Two major haplogroups were identified in common wheat: Chinese Spring (CS) and LM15 groups (Supplementary Data 7). We then compared *MS2* in nine accessions (Supplementary Fig. 12; Supplementary Data 3). In the CS haplogroup, *Ms2* has a *Taigu* retroelement (inserted at nucleotide position −314..−310), but the male-fertile *ms2* does not; the only difference between these two alleles is the *Taigu* retroelement. *Taigu* has two identical LTRs (direct repeats, 582 bp), suggesting a recent formation of *Ms2* within the CS lineage. There were seven exonic SNPs between CS and LM15; $SNP_{3404}$ caused a premature stop codon (L284*) in the LM15 group. Both *Ae. tauschii* and common wheat had two additional variants: one in the promoter (at nucleotide position −673..−364) and one in the second intron (at nucleotide position 902..905).

We further genotyped 575 accessions using four haplotype markers, which target polymorphic sites located at nucleotide positions −673..−364, −314..−310, 902..905 and 3404 (Table 1; Supplementary Data 3, 4 and 8). The *MS2* gene is associated with at least 18 haplotypes, three in the CS group (A1-A3; 71 lines), five in the LM15 group (B1-B5; 451 lines), seven in the hybrid group (H1-H7; 24 lines), and three in the null

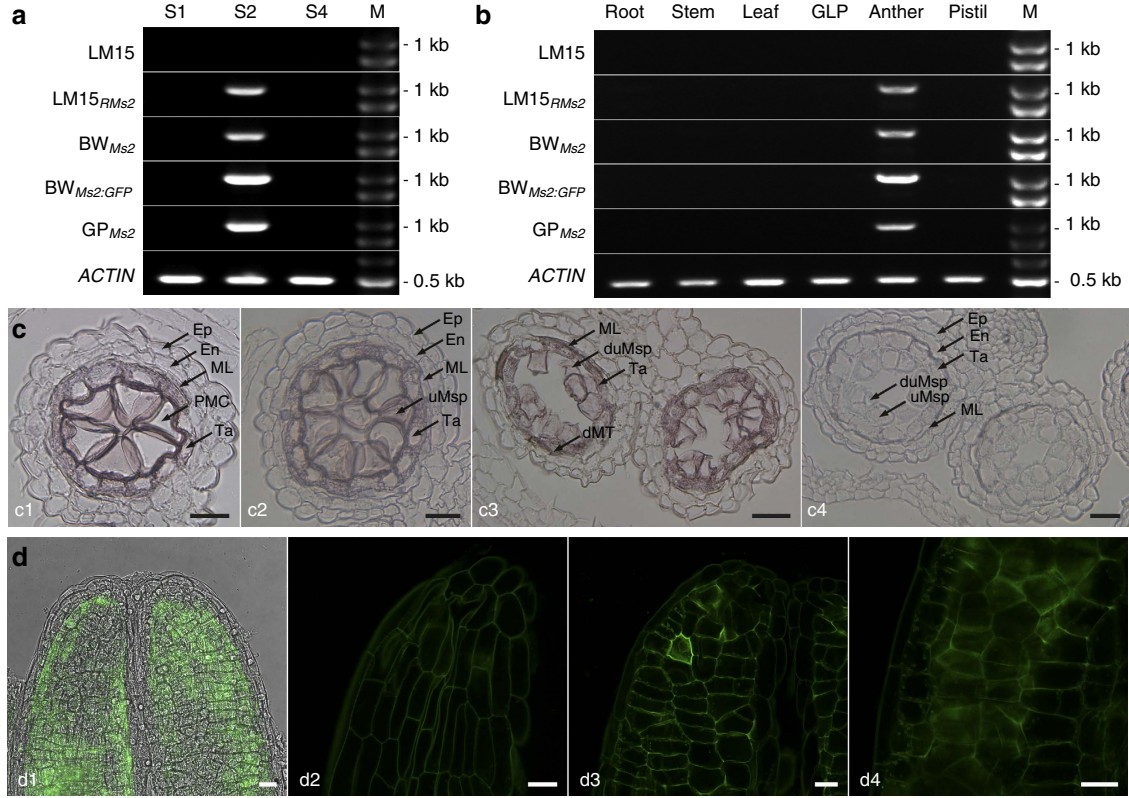

**Figure 4 | Spatiotemporal expression of *Ms2*.** (**a**,**b**) *Ms2* expression in anthers. BW$_{Ms2}$, BW$_{Ms2:GFP}$, and GP$_{Ms2}$ are male-sterile transgenic plants JZ7-2_2, JZ19-6, and 22-1A, respectively. GLP was pooled from glumes, lemma and palea. (**c**) *In situ* hybridization of *Ms2* mRNA in cross sections of LM15$_{RMs2}$ stamens. Hybridization with antisense probes was detected in pollen mother cells (PMC, c1), early uninucleate microspores (c2), and late uninucleate microspores (c3); hybridization with sense probes was negative in uninucleate microspores (c4). (**d**) Ms2:GFP in optical sections of an undissected male-sterile anther (from JZ19-6) at the PMC stage. (d1) A superimposed laser scanning confocal and light micrograph showing GFP across the end of an anther. GFP was detected in the middle layer (d2), tapetum (d3) and PMC (d4). The experiment was repeated three times. En, endothecium; Ep, epidermis; ML, middle layer; Ta, tapetum; uMsp, uninucleate microspore; dMT, degenerated middle layer and tapetum; duMsp, degenerated uninucleate microspore; M, the 250 bp DNA Ladder (GENEray, Shanghai, China). Scale bars, 25 μm (**c**) and 20 μm (**d**).

group (N1-N3; 29 lines) (Table 1; Supplementary Data 8). Overall, 74% and 12% of our assessions are in haplotypes B1 in the LM15 and A1 in the CS groups, respectively. Common wheat and *Ae. tauschii*, contain both A1 and B1 haplotypes, suggesting these haplotypes diverged before the formation of common wheat, and were inherited independently in common wheat. Sixty-eight per cent of our *Ae. tauschii* accessions had SNP3404. All the remaining haplotypes are unique to either *Ae. tauschii* or common wheat, and apparently evolved from the major haplotypes via mutations, recombinations and/or deletions. All null group accessions are in *Ae. tauschii*, illustrating a common phenomenon of partial or complete deletion of *MS2* DNA in the D-genome ancestor.

Of the four regions investigated, we compared the number of accessions in three genotype groups (A, B, and a 'non-AB' group, Table 1) of all materials except for those in the null group ($n = 546$). A $\chi^2$ revealed no significant difference among HT1, HT3 and HT4, but HT2 was associated with a higher number of 'non-AB' accessions than those in either HT1, HT3 or HT4 ($\chi^2 > 35.2$, df $= 2$, $P < 2.2 \times 10^{-8}$). Therefore, the polymorphic site $-314..-310$ (HT2) is the most variable and might be a region that is more likely to be involved in genetic change. We conclude that *Ms2* (haplotype A2) was formed when the retrotransposon *Taigu* inserted into the A1 haplotype.

Here, we propose that a *Ms2* allele (*Ms2*$_{O261S}$) originated in 'O261,' a line in PopA that has an intragenic recombination in *MS2*. *Ms2*$_{O261S}$ inherited its promoter and first two exons from the dominant *Ms2* in XY6$_{Ms2}$, but the rest of the gene comes from a recessive *ms2* in SW41 (Supplementary Data 3). Despite having a large portion of the recessive *ms2* allele and the T3404A mutation, O261 is sterile, which suggests that the hybrid *Ms2*$_{O261S}$ gene confers male sterility and that nucleotide differences between MS and MF alleles in the third to last exons have no effect on *MS2* function. Therefore, we propose that the retroelement insertion acts as a molecular switch that turns on *MS2* expression and leads to functionalization of the gene.

**Putative interaction network of *Ms2*.** To reveal the effect of *Ms2* on the transcriptome, we compared the S2-stage mRNA from anthers of LM15 and LM15$_{RMs2}$, using flag leaves and pistils as controls. Based on the wheat genome annotation (IWGSC1.0 + popseq.28), we detected 72,499 active genes in anthers, leaves, and/or pistils (Supplementary Fig. 13a). After removing minimally-expressed genes, we extracted 7,294 differently expressed genes (DEGs) between fertile and sterile anthers (adjusted *P* value $< 0.01$). Using higher stringency, we then extracted 2,991 sterile-anther-enriched genes (SAEGs) and

939 fertile-anther-enriched genes (FAEGs) (Supplementary Fig. 13b). Among them, 1,907 SAEGs were in the gene ontology (GO) database and had 127 significant GO terms, and 537 FAEGs had 59 significant GO terms (Supplementary Data 9).

In the FAEG group, genes with the GO terms 'metabolism' (GO:0008152, $P = 4 \times 10^{-10}$; hereafter in this paragraph: the $P$ value is based on the Fisher's statistical test, and a FDR-adjusted $P$ value is also indicated in the Supplementary Data 9), 'photosynthesis' (GO:0015979, $P = 4 \times 10^{-76}$), and 'translation' (GO:0006412, $P = 5 \times 10^{-14}$) were overrepresented. Among molecular function GO terms, the 'structural constituent of ribosome' was also significantly overrepresented in the FAEG group (GO:0003735, $P = 2 \times 10^{-20}$). The cellular component GO terms 'macromolecular complex', 'membrane', 'thylakoid', 'ribosome' and 'intracellular non-membrane-bounded organelle' were overrepresented. In the SAEG group, there were also many genes involved in metabolism (GO:0008152, $P = 2 \times 10^{-16}$), oxidation reduction (GO:0055114, $P = 9 \times 10^{-13}$), and protein modification (GO:0006464, $P = 5 \times 10^{-21}$). Other GO terms overrepresented among SAEG genes include 'cell wall organiza-tion' (GO:0071554, $P = 3 \times 10^{-10}$) and 'cell wall macromolecule catabolic process' (GO:0016998, $P = 9 \times 10^{-16}$). Among molecular function GO terms, 'catalytic activity' (GO:0003824, $P = 3 \times 10^{-26}$) was overrepresented and genes predicted to encode oxidoreductase, transferase, and chitinase were among this group. 'Extracellular region' (GO:0005576, $P = 2 \times 10^{-6}$) was the only significantly overrepresented cellular component GO term among SAEG genes.

We screened for MS2-interacting proteins in the S2-stage spikes using yeast two-hybrids (Y2H) analysis, and identified 312 positive clones encoding for 142 proteins (Supplementary Fig. 13c,d; Supplementary Data 10). Analysis indicated four significant GO terms 'nucleotide binding' (GO:0000166, $P = 0.00001$; hereafter in this paragraph: the $P$ value is based on the Central Limit Theorem, and a FDR-adjusted $P$ value is also indicated in the Supplementary Data 9), 'purine nucleotide binding' (GO:0017076, $P = 0.00001$), 'ribonucleotide binding' (GO:0032553, $P = 0.00001$), and 'purine ribonucleotide binding' (GO:0032555, $P = 0.00001$) for 17 entries (Supplementary Fig. 13e; Supplementary Data 9 and 10), including eight for the eukaryotic elongation factor (eEF1A), which accounted for 25% of the 312 positive clones, the eukaryotic initiation factor (eIF-4A), and the GTP-binding protein (GTPase) (Supplementary Data 10). Among the other interacting proteins, there were two eukaryotic elongation factors (eEF1B) and seven ribosomal proteins. Using pairwise Y2H, we then demonstrated that wild-type MS2 appears to form homodimers and hetrodimers, and that wild-type MS2 interacts with full-length proteins of interest (Supplementary Figs 13f and 14).

## Discussion

Common wheat is a hexaploid species, which has 124,201 annotated genes, 80% repetitive sequences, and a 17-gigabase haploid genome[27]. Consequently, map-based positional cloning in common wheat is complicated and time consuming. There are three dominant GMS genes reported in wheat: *Ms2*, *Ms3*, and *Ms4* (ref. 13); *Ms2* is on chromosome arm 4DS[14]. The wheat D genome (originally from *Ae. tauschii*) has low DNA diversity, which complicated positional cloning of genes in this genome. Synthetic hexaploid wheat (derived from *Ae. tauschii*) has considerable diversity when compared to the D genome in common wheat. However, the *MS2* region is partially or completely deleted in 40% of *Ae. tauschii* accessions tested here. Consequently, the particular selection of synthetic

wheat was critical for the cloning of the *Ms2* gene. For example, the deletion of the *Ms2* region (*X25-Xsdauw30*, Supplementary Data 7) in the synthetic wheat SW7 impeded further mapping of *MS2* within the *Xsdauw20–Xsdauw32* interval. The *RMs2* linkage has a tandem segmental duplication over a 1 Mb region[28], which apparently was responsible for our exceptionally low recombination in the *Ms2* region among the *RMs2*-based populations (popE and popF, Supplementary Fig. 4). Regardless, low recombination complicates positional cloning, as indicated with the male-sterile gene *Ms3* (ref. 29). By selecting the regular Taigu line XY6$_{Ms2}$ and the synthetic wheat SW41 as parents, we started with sufficient variation in the D genomes and produced progeny with recombination in the *MS2* region, which facilitated successful cloning of the *Ms2* gene.

*Taigu_P1593-1* (*Taigu*) represents a new family of non-autonomous LTR retrotransposon. First, *Taigu_P1593-1* is a TRIM retroelement[30]. *Taigu_P1593-1* is associated with two identical LTRs and a perfect TSD, suggesting a recent origin of this retroelement. Second, the *Taigu* retroelements only emerged in the Triticeae lineage; there are 50 repetitive sequences in common wheat and 28 repetitive sequences in *Ae. tauschii* that are similar to part or to a complete *Taigu* element. Third, there are no significant hits of *Taigu* in the Triticeae repeat sequence database[31]. The insertion of *Taigu* into the *MS2* promoter results in *Ms2* expression in anthers of Taigu lines. There is one somewhat analagous case in *Citrus sinensis*[32]. We do not know whether the *Taigu* retroelement acts as a transcription enhancer or its insertion destroys a repressive *cis*-element. In LTR-type retrotransposons, the 5′ LTR usually contains a promoter for producing template RNAs[30]. The retrotransposon/LTR-based promoters can directly induce gene expression[32] or can play a role in activating 'sleeping' genes[33]. LTR retrotransposons also can have finely tuned responses to a diverse array of external stimuli, and can act as dispersed regulatory modules that can respond to external stimuli and activate adjacent plant genes[34]. If there were a repressive *cis*-element in the promoter region of the *MS2* gene, other sequence variants of the *cis*-element would likely also activate the *MS2* gene. For example, the H2 and H3 haplotypes (e.g., 'CIae 30' and 'AL8/78', Supplementary Data 8) are associated with a 49-bp deletion in close vicinity (7 bp) to the insertion site of the *Taigu* retroelement, but accessions of the H2 and H3 haplotypes are male fertile. Consequently, we hypothesize that the *Taigu* retroelement acts as a transcription enhancer and activates the transcription of the *MS2* gene.

In China, the *Ms2* gene has been widely used during recurrent selection in conventional breeding of wheat[35]. The cloning of the dominant *Ms2* could boost recurrent selection in wheat globally and enable additional applications for population improvement and gene pyramiding in autogamous crops[22]. For instance, any transformation-amenable wheat can be engineered to carry the *Ms2* gene, which will facilitate a *Ms2*-based recurrent selection in the 'engineered Taigu' (eTaigu) line (Supplementary Fig. 15). Although eTaigu lines would be classified as either cisgenic or transgenic, the final cultivars would have male fertility restored and be *Ms2*-free. Consequently, use of the *Ms2* gene may be more readily accepted by the public. Here, we also demonstrated that *Ms2* confers male sterility in barley and *Brachypodium*, in addition to wheat. The range of crops in which the transgenic *Ms2* could function is unknown. Nonetheless, the potential for cereal and perhaps other crop improvement is great.

Hybrid corn and rice cultivars are renowned for their higher yield and other superior traits compared to inbreed lines. Despite interest by wheat breeders in hybrids, the development of wheat hybrids globally has been limited by a lack of tools. Cloned *Ms2* offers a straightforward method to use cisgenics or transgenics

transiently in either wheat or barley breeding programs. Here, we propose a high-throughput hybrid production system using any wheat that can be transformed with *Ms2* plus an aleurone-specific gene for pigmented kernels (MSC for *Ms2* colour wheat) (Supplementary Fig. 16). The MSC system has two potential applications: (1) screening for desirable heterosis groups, and (2) hybrid seed production for either specific heterosis groups or, for example, for plants with new combinations of disease resistance genes.

This work sheds light on plant male sterility. In wild-type anthers, the middle layer and tapetum (Supplementary Fig. 1) eventually degenerate via programmed cell death[36], but not until after the pollen mother spores undergo meiosis. In Taigu lines, the middle layer starts to degenerate as early as the sporogenous cell stage, and any uninucleate microspores that are produced collapse; the collapse is correlated with the expression of *Ms2* in the middle layer and the male reproductive cells.

In conclusion, we report molecular characterization of a male sterility gene in wheat. The current *Ms2* gene has played pivotal roles in wheat breeding, enabling the release of hundreds of breeding lines/cultivars in common wheat. Remarkably, *Ms2* homologues are only found in wheat and its close relatives, and are absent from other crops outside the Triticeae tribe. The cloned *Ms2* gene provides high value for breeding and producing hybrid wheat, and potentially for other major crops.

## Methods

**Plant materials and growth conditions.** We used 18 plant accessions and four BC$_1$F$_1$ populations, including those of common wheat (*Triticum aestivum* L., $2n = 6x = 42$) and synthetic hexaploid wheat (SHW, $2n = 6x = 42$) (Supplementary Table 1). Additional germplasm was used for *MS2* diversity (Supplementary Data 8). Plants were grown in the greenhouse with a 16 h photoperiod and a daytime and nighttime temperature of 22–25 °C and 15–20 °C, respectively, in the growth chamber with a 16 h photoperiod and a constant temperature of 23 °C, or in the research farm of the Shandong Agricultural University, Taian, China.

'Langdon', 'LDN$_{DIC521-2B}$', and three SHW lines (SW7, SW8 and SW41) were received from Dr J.D. Faris at the USDA-ARS, Fargo, ND, USA. The remaining SHW lines were provided by Dr T. Payne at the International Maize and Wheat Improvement Center, El Batán, Texcoco, México. All *Ae. tauschii* accessions except for 'AL8/78' were provided by Dr H.E. Bockelman at the USDA-ARS, Aberdeen, ID, USA.

**Anther development in common wheat.** We inspected wheat anthers at four stages: when the auricle distance between the flag and penultimate leaves was from −3 to −5 cm (S1 ≈ a Zadoks scale Z38, ref. 37); when the auricle distance was from 3 to 5 cm (S2 ≈ Z41); when the first awn was visible in an undissected spike (S3 ≈ Z49); and when all of the first inflorescence was visible (S4 ≈ Z59). Anthers and pollen were either stained directly using the Alexander protocol[38] which dyes viable pollen either red or pink, or embedded using a modified protocol for semi-thin sections[39]. For semi-thin sections, anthers were fixed in glutaraldehyde (3% glutaraldehyde in 0.2 M phosphate buffer, pH 7.4), dehydrated using an increasing gradient of ethanol (30–100%), cleared using an increasing gradient of xylene (25–100% in ethanol), infiltrated with paraffin, and cut into 8 μm thick sections. Anther sections were then dewaxed, rehydrated, stained in 0.02% ruthenium red and dehydrated using an increasing gradient of ethanol (30–100%). The dehydrated anther sections were washed with an increasing gradient of xylene (50–100% in ethanol), air-dried, and mounted in neutral balsam for future use. Anther, pollen and their sections were examined under an Olympus SZX16 stereo microscope (Olympus, Shinjuku, Tokyo, Japan) or a Nikon ECLIPSE Ni microscope (Nikon, Shinagawa, Tokyo, Japan). Images were recorded with a Digital Sight DS-Fi camera (Nikon), and processed using the NIS Elements 4.0 (Nikon).

**Polymorphic status among parental lines.** We used the wheat 90 K iSelect SNP array[40] to genotype six hexaploid lines (CS$_{RMs2}$, LM15$_{RMs2}$, XY6$_{Ms2}$, SW7, SW8 and SW41; genome = AABBDD), two tetraploid wheat (Langdon and LDN$_{DIC521-2B}$; genome = AABB), and two diploid accessions (CIae 9 and PI 569536; genome = DD). To extract the A and B-specific SNP markers, we discarded non-specific or low quality SNP markers by performing a three-step elimination: (1) those with any missing data or any heterozygous genotypes within Langdon or LDN$_{DIC521-2B}$; (2) those with any genotype data within CIae 9 or PI 569536; and (3) those with any missing data or any heterozygous genotypes within the six parental lines. We used a similar procedure to extract the genome D-specific SNP

markers, but step one was for CIae 9 or PI 569536 and step two was for Langdon or LDN$_{DIC521-2B}$. Using genome-specific SNP markers (5,975 in AB and 1,563 in D), we calculated the proportion of polymorphic SNP markers for 15 pairwise comparisons between the parental lines.

**Precise mapping of the *Ms2* gene.** To map the *Ms2* gene, we used three Taigu lines (CS$_{RMs2}$, LM15$_{RMs2}$ and XY6$_{Ms2}$) and three SHW lines (SW7, SW8 and SW41) (Supplementary Table 1). The three SHW lines share the same durum donor[41]. We started of a 3 × 3 factorial backcross between Taigu and SHW lines. The same SHW line was used as the recurrent male parent for each backcross. In total, we prepared eight BC$_1$F$_1$ populations; the XY6$_{Ms2}$/SW8 failed to produce seed.

We initially developed nine PCR markers using the *Ae. tauschii* map[24], and mapped the *Ms2* gene to the *Xsdauw8-Xsdauw36* interval, which aligns to the AT4D3406-AT4D3410 region in *Ae. tauschii* (Supplementary Fig. 4). AT4D3406 and AT4D3410 anchor to the physical contigs ctg16527 and ctg10366, respectively, of which ctg10366 appears to span most of the AT4D3406-AT4D3410 interval (Supplementary Fig. 5). We selected overlapping BAC clones for sequencing and marker development. As a result, we designed a distal marker *Xsdauw11* on RI298G16, and a proximal marker *Xsdauw35* on MI225K13 (Supplementary Figs 4 and 5). Coincidently, both *Xsdauw11* and *Xsdauw35* anchor to the physical contig ctg10366, defining a physical map of *MS2*. We then used 13 overlapping BACs (Supplementary Fig. 5) to narrow down the *MS2* interval. During the process, we developed 19 markers in the *Xsdauw11-Xsdauw35* interval and precisely mapped the *MS2* gene between *Xsdauw27* and *Xsdauw29*. Genetic maps were constructed using the regression algorithm in JoinMap 4.0 (Kyazma B.V., Wageningen, Netherlands). The recombination frequency was transformed into centimorgans (cM) using the Kosambi function.

**Physical map of the *Ms2* region in Taigu.** Using standard prototols[42,43], we constructed a BAC library of LM15$_{RMs2}$. Briefly, high-molecular weight genomic DNA was extracted from leaf tissues, partially digested by the restriction enzyme *Hind*III, and ligated into the BAC vector pIndigoBAC536-S[43]. The ligation product was transformed into the ElectroMAX DH10B T1 phage-resistant cells (Invitrogene, Carlsbad, CA, USA) and screened on the LB medium with 12.5 mg l$^{-1}$ chloramphenicol, 80 mg l$^{-1}$ X-gal and 100 mg l$^{-1}$ IPTG. White colonies were individually picked into 384-well microtiter plates. In total, 706,176 BAC clones were arranged in 1,839 384-well plates, representing a 5.1-fold coverage of the wheat genome (~17 Gb) (Supplementary Table 2).

We developed a PCR-based screening of the LM15$_{RMs2}$ BAC library. In brief, a 384-well stock was duplicated and pooled to extract a primary DNA pool using the ZR BAC DNA Miniprep Kit (Zymo Research Corporation, Irvine, CA, USA). A super DNA pool was then prepared by mixing ten primary DNA pools. In total, we prepared 1,839 primary DNA pools and 184 super DNA pools. We screened the BAC library using *Xsdauw11*, *Xsdauw20*, *Xsdauw25*, *Xsdauw33* and *Xsdauw35*, and recovered twelve BAC clones in this region (Supplementary Fig. 6). We then used *Xsdauw26* and *X26* to determine whether these BAC clones were derived from a *ms2* or *Ms2*-associated chromosome.

High-throughput sequencing was done by the Berry Genomics Company (Beijing, China). The BAC DNA was processed into a paired-end (PE) DNA library. In brief, DNA was acoustically fragmented using the Covaris instrument (Covaris, MA, USA), end-repaired and 3′ adenylated using the NEBNext Sample Reagent Set (New England Biolabs, Ipswich, MA, USA), ligated to Illumina adaptors, and separated on a 2% agarose gel to select fragments about 400–500 bp. Adaptor specific primers were used to amplify the ligation products. The final library was evaluated by quantitative RT-PCR with the StepOne Plus Real-Time PCR system (Applied Biosystems, Foster City, CA, USA). PE reads (150 bp) were obtained using the Illumina HiSeq2500 (Illumina, San Diego, CA, USA).

Using raw reads, we eliminated adaptors, low quality PE reads (half or more bases of any PE reads with a quality value $Q \leq 5$ or unknown bases accounting for over 10% of any PE reads), and then removed any vector or *E. coli.* sequence using the cross_match tool in Phrap[44]. We then performed a *de novo* assembly on clean reads (*ca.* 4.5 Gb in total for all BACs) using the ABySS 1.5.2 programme[45]. Specific BAC clones were further assembled to cover the *Ms2* and *ms2* regions. In the *Xsdauw24-Xsdauw32* interval, we identified several *Ms2* candidates and compared their transcription in Taigu and transgenic plants (Figs 2c and 4a,b; Supplementary Figs 9a, 17 and 18).

***PG5$_{P1593S}$* and *PG5$_{P1076F}$* transcripts in LM15$_{RMs2}$.** We estimated the relative expression of *PG5$_{P1593S}$* and *PG5$_{P1076F}$* in LM15$_{RMs2}$ by comparing their frequency in 5′- and 3′-terminal cDNA clones, internal cDNA clones and RNA-Seq reads.

For 5′- and 3′-terminal cDNA clones, we synthesized the S2-stage spike cDNA of LM15$_{RMs2}$ using the RevertAid Frist Strand cDNA Synthesis Kit (Thermo Scientific, Waltham, MA, USA), and conducted the rapid amplification of cDNA ends (RACE) to recover the 5′ and 3′ untranslated-regions of *PG5* using the SMARTer RACE cDNA Amplification Kit (Clontech, Mountain View, CA, USA). The 5′ RACE utilized the *PG5*-specific primers P100 and P101, as well as the 3′ RACE with P102 and P103 (Supplementary Data 2). PCR products were cloned into the pMD18-T vector (Takara, Dalian, China) and sequenced individually. For

internal cDNA clones, we first amplified the *PG5* cDNA using the conserved primers P104 and P100 (Supplementary Data 2), and cloned the PCR products into the pMD18-T vector. Each PCR clone was again amplified using P104 and P100, and was differentiated by a cleavage amplification polymorphism sequence (CAPS) that uses the restriction enzyme *TaqI* (New England Biolabs). PCR products of the $PG5_{P1593S}$ clone were cut into two bands (398 bp and 42 bp), but only one band (440 bp) in the $PG5_{P1076F}$ clone, which was scored in a 1.5% Agrose gel (Supplementary Fig. 18). Specific RNA-seq reads of $PG5_{P1593S}$ and $PG5_{P1076F}$ were identified as described in the Method section: RNA sequencing and GO analysis.

**Mutagenesis and mutation screening.** We used backcrossed seeds of LM15$_{RMs2}$/LM15 (the LM15$_{RMs2}$ set) and the BC$_1$F$_1$ seeds of CS$_{RMs2}$/2*SW7 and CS$_{RMs2}$/2*SW8 (the CS$_{RMs2}$ set) for chemical mutagenesis. Using ethyl methane sulfonate (EMS), we prepared the mutant populations. Briefly, lots of 400 seeds (M$_0$) were soaked in 100 ml ethyl methane sulfonate (EMS, Sigma-Aldrich, St Louis, MO, USA) solution (87.4 μM in water), incubated on a shaker at 150 r.p.m. at 25 °C for 10 h, and washed with running water at room temperatures for 4 h. After germination, vigorous seedlings with roots were grown in greenhouse and field conditions. Because the populations of mutagenized seeds had an expected 1:1 segregation of dwarf, male-sterile *RMs2*:wild-type male-fertile *rms2*, at the flag-leaf stage, we screened the M$_1$ plants for a dwarf phenotype (≤60 cm) and the presence of a $PG5_{P1593S}$ allele using the HT5 marker (Supplementary Data 4) and discarded those that were tall (>60 cm) and had no $PG5_{P1593S}$. Twelve hundred plants in the LM15$_{RMs2}$ set and 1,066 plants in the CS$_{RMs2}$ set were retained for mutation screening.

For the LM15$_{RMs2}$ set, genomic DNA was prepared from the main-stem flag leaf using the Sarkosyl method[46]. The DNA samples were pooled four-fold and organized into a 96-well format. Flag-leaf DNA was also prepared from a main stem or tiller that produced a male-fertile spike; samples were pooled one-fold with the genomic DNA of LM15$_{RMs2}$. For the CS$_{RMs2}$ set, the flag-leaf DNA was only prepared from a main stem or tiller that produced a male-fertile spike, and samples were pooled one-fold with the genomic DNA of CS$_{RMs2}$.

Positive DNA pools with a mutation in the $PG5_{P1593S}$ gene were identified using a modified TILLING approach[47]. We first screened for positive DNA pools, which involved two PCR reactions. A long-range PCR was performed to amplify a 5,925-bp fragment (between primers P133 and P138; Supplementary Data 2) by using the KOD FX kit (Toyobo Co., Osaka, Japan). The PCR product was diluted 500 times using ddH$_2$O and then used as a template to amplify the following three regions: (1) exons 1–2 between P133 and P134, (2) exons 3–4 between P135 and P136 and (3) exons 6–8 between P137 and P138 (Supplementary Data 2). PCR products were then subjected to a slow denaturing and re-annealing to enhance heteroduplex formation. The PCR heteroduplexes were digested with celery juice extract (CJE)[48]. Positive DNA pools were those associated with one or more specifically cleaved PCR products (Supplementary Fig. 8). For two-fold pools, a positive pool directly anchored to a specific DNA sample. For four-fold pools, each DNA of a positive pool was mixed with LM15$_{RMs2}$ to prepare a two-fold DNA pool, which was further used to identify a mutant. Specific M$_1$ mutants with a mutation in the $PG5_{P1593S}$ gene were selected (Supplementary Fig. 8). The base change was identified by sequencing a specific PCR product in M$_1$ or M$_2$ plants and was confirmed using PCR markers (Supplementary Figs 8 and 17; Supplementary Data 4 and 5).

**Plant genetic transformation.** For genetic transformation, we prepared four types of plasmids: (1) native expression of *PG5*, (2) native expression of *PG5:GFP*, (3) overexpression of *PG5* and (4) overexpression of *PG5:GFP*. For native expression, we used a 10,592-bp genomic fragment ( −5578..4078..+936) of the $PG5_{P1593S}$ gene, which includes the native promoter (NP1, 5,578 bp), the genomic coding region (4,078 bp) and the native terminator (NT, 936 bp). We isolated a 10,592-bp fragment from LM15$_{RMs2}$ (or BAC clones) using PCR primers P139 and P140 (Supplementary Data 2), and assembled the plant expression construct PC976 (NP1::$PG5_{P1593S}$) (Supplementary Table 3). By using the first 9,653 bp that lacks the stop codon and NT, we then assembled the tag construct PC973 (NP1::$PG5_{P1593S}$:GFP) (Supplementary Table 3).The KOD FX kit was used for PCR amplification. To overexpress the *PG5* gene, we prepared the $PG5_{P1593S}$- and $PG5_{P1076F}$-type cDNA from LM15$_{RMs2}$ using the PCR primers P149 to P151 (Supplementary Data 2), and assembled three plant expression constructs: PC970 (Ubi::$PG5_{P1593S}$), PC971 (Ubi::$PG5_{P1076F}$), and PC972 (Ubi::$PG5_{P1593S}$:GFP) (Supplementary Table 3). These binary constructs have *BAR* and hygromycin selection markers on their T-DNA. Plasmid DNA was used for biolistic bombardment in wheat. For *Agrobacterium*-mediated transformation of barley and *Brachypodium*, we introduced the binary constructs into the *Agrobacterium* strain AGL1.

Using standard methods on plant transformation and tissue culture, we performed biolistic bombardment in wheat[49,50] and *Agrobacterium*-mediated transformation in barley[51] and *Brachypodium*[52]. For PC976, we bombarded 2,742 immature embryos of wheat 'Bobwhite', infected 500 immature embryos of 'Golden Promise' and 100 immature embryos of 'Bd21-3' using AGL1 cells harbouring PC976, and generated putative transgenic plants (Supplementary Data 6). For PC973, we bombarded 2,102 immature embryos of Bobwhite and

generated putative transgenic plants (Supplementary Data 6). For overexpression, we infected 500 immature embryos of Golden Promise and 200 immature embryos of Bd21-3 with each of three constructs (PC970, PC971 and PC972), bombarded 1,374 immature embryos of Bobwhite for PC972, and generated putative transgenic plants (Supplementary Data 6).

Regular PCR was used to confirm transgene integration: P121 and P122 for *BAR*, P133 and P141 for $PG5_{P1593S}$ and $PG5_{P1593S}$:*GFP* (Supplementary Figs 9 and 17; Supplementary Data 2). For the $PG5_{P1593S}$ positive transgenic plants, a 36-cycle RT-PCR was used to confirm transcription of the transgene and the *ACTIN* control; we used PCR primers P129 and P130 for $PG5_{P1593S}$, P129 and P151 (or P152) for $PG5_{P1593S}$:*GFP*, P142 and P143 for *ACTIN* (Supplementary Data 2). The same *ACTIN* primers were used for both wheat, barley and *Brachypodium*, but only those from LM15$_{RMs2}$ are shown in Fig. 4 and Supplementary Fig. 17.

**Cellular localization of the MS2 protein.** To study cellular localization, we used four constructs: PC972 (Ubi::*Ms2*:GFP) and PC983 (35S::GFP:*Ms2*), and their respective controls PC134 (Ubi::GFP) and pMDC43 (35S::GFP) (Supplementary Table 3). Each construct was stored in *Agrobacterium* strain AGL1. We performed transient expression using the tobacco relative, *Nicotiana benthamiana* Domin. Four to six week old tobacco plants were co-infiltrated with an equal mixture of two *Agrobacterium* suspensions, one with the target construct and one with the 35S-driven p19 suppressor[53]; suspensions were made using 10 mM MES (pH 5.7), 10 mM MgCl$_2$ and 100 μM acetosyringone. To prepare mesophyll cells, we removed the abaxial epidermis of the infected leaf tissue (3 mm × 10 mm), and soaked the remaining tissue in a fresh enzyme solution (0.6 M mannitol, 1.3% cellulase, 0.75% macerozyme, 20 mM MES (pH 5.7), 0.1% BSA and 2 mM CaCl$_2$; Sangon Biotech, Shanghai, China) for 2 h at room temperature. The protoplast suspension was inspected under a Leica TCS SP5 II laser scanning confocal microscope (Leica Microsystems, Wetzlar, Hesse, Germany). Three views were captured for each target cell: (1) a brightfield, (2) a GFP channel and (3) a chlorophyll autofluorescence (ChlAF).

**RNA sequencing and GO analysis.** We collected four types of samples from plants grown in greenhouses to the S2 stage: fertile anthers of LM15, sterile anthers of LM15$_{RMs2}$, pistils of LM15 and LM15$_{RMs2}$, and flag leaves of LM15 and LM15$_{RMs2}$. Tissues were stored in liquid nitrogen during collection and then at −80 °C for storage. There were three biological replicates per tissue-genotype.

We extracted total RNA using TRIzol (Invitrogen) and evaluated the RNA integrity with an Agilent 2100 Bioanalyzer (Agilent Technologies, Palo Alto, CA, USA). The Berry Genomics Company prepared the sequencing libraries (*ca.* 500 bp per insert) and performed high-throughput sequencing (125 bp PE reads) using HiSeq2500. We pre-processed raw reads using the FastQC algorithm (http://www.bioinformatics.babraham.ac.uk/projects/fastqc/), and performed quality trimming and adaptor removal by Trimmomatic[54]. In total, we retrieved 484.25 million clean pairs (Supplementary Table 5), and used them for mapping and transcript assembly. For the *PG5* gene, specific reads were indicated by a perfect match to $PG5_{P1593S}$ or $PG5_{P1076F}$ of LM15$_{RMs2}$ using the STAR programme[55] and a customized Perl script.

On the basis of the current wheat genome[27], clean PE reads were mapped to a repeat-masked data set using the STAR programme[55]. The PE reads were divided into three groups: (1) unique mapped reads—those mapped to a unique location in the masked genome, (2) multiple mapped reads—those mapped to at least two locations in the masked genome, (3) unmapped reads—those absent in the masked genome or those mapped to repeats (Supplementary Table 6). Unique mapped reads were retrieved from the STAR-derived BAM files using Perl scripts.

Based on the wheat genome (IWGSC1.0 + popseq.28, http://plants.ensembl.org), we assembled the unique mapped reads using htseq-count script (v.0.6.1) with the union mode in the HTSeq package[56], and quantified their normalized transcript counts using the DESeq package (v.1.26.0)[57]. There were 79,153 genes with at least one nonzero count value in the four types of samples. To remove false positives and poorly-expressed genes[27], we defined a count cutoff value of 1.06 (Supplementary Table 7). The count values less than 1.06 were assigned a zero value; 72,499 expressed genes were retained (Supplementary Table 7). Between male-fertile (MF) and male-sterile (MS) anthers, we determined 7,294 differentially expressed genes (DEGs) by controlling the false discovery rate (FDR<0.01) for multiple testing in DESeq (v.1.26.0)[57] and using the ajdusted *P* value ($P_{adj}$ <0.01). Furthermore, we divided the anther-related DEGs into two groups: (1) sterile-anther-enriched genes (SAEG): those associated with higher transcription in sterile anthers compared to the fertile anthers; (2) fertile-anther-enriched genes (FAEG): those associated with higher transcription in fertile anthers compared to the sterile anthers. The following rules were used to define the final list in each group: (1) genes active in only one type of anther with a count value >the 10$^{th}$ percentile (3.29) of the cleaned data (Supplementary Table 7); (2) genes active in both MF and MS anthers but the relative amount was >10 fold in either MF or MS anthers. By this way, we extracted 2,991 and 939 genes in the final SAEG and FAEG lists, respectively.

The 3,930 anther-enriched DEGs were analysed for GO frequencies. All genes were first assigned identities with the agriGO toolkit[58], and then compared to the *Triticum aestivum* transcript (v2.2) using the singular enrichment analysis (SEA). To prepare heat maps of the anther-enriched DEGs, their expression values were first lg transformed (normalized count  +1), and then organized in an expression

matrix. A hclust() function and 'average' method in R was used to cluster the expression matrix using the correlation distance. We then created heatmaps using the Heatmap.2 of the R package gplots (http://www.inside-r.org/packages/cran/gplots/docs/heatmap.2).

**Yeast two-hybrid analysis.** Both *Ms2* ( = $PG5_{P1593S}$) and *ms2* ( = $PG5_{P1076F}$) were targeted for Y2H analyses. The coding sequence of each gene was amplified from LM15$_{RMs2}$ using PCR primers P153 and P150 (Supplementary Data 2), cloned into the pENTR/D-TOP vector (Invitrogen), and recombined into the Gateway-compatible pGADT7 (AD, prey) and pGBKT7 (BD, bait) vectors[59]. Four Y2H constructs were prepared including PC986 (*ADH1::AD:Ms2*) and PC987 (*ADH1::BD:Ms2*), PC988 (*ADH1::AD:ms2*), and PC989 (*ADH1::BD:ms2*) (Supplementary Table 3). Similarly, we prepared the AD and/or BD plasmids for four $PG5_{P1593S}$ cDNA variants and eight MS2-interacting protein genes (Supplementary Table 3).

The Y2H assay included the use of the double dropout medium (DDO: SD/-Trp/-Leu), the triple dropout medium (TDO: SD/-Trp/-Leu/-His), and the quadruple dropout medium (QDO: SD/-Trp/-Leu/-His/-Ade) (Clontech). We first performed autoactivity tests (Supplementary Fig. 14). We then constructed a cDNA library and performed the Y2H screening using standard methods[60]. The cDNA library was prepared on the S2-stage spikes of LM15$_{RMs2}$. Aliquots of 1 ml with a titre value of $9 \times 10^7$ c.f.u. ml$^{-1}$ were prepared and stored in a $-80\,°C$ freezer. Both Ms2 and ms2, as bait proteins, were used to screen the cDNA library on the QDO and TDO media, respectively. Positive clones from the library screening were tested for their autoactivity and pairwise interactions with Ms2 and ms2. Selected clones were sequenced using a T7 primer (Supplementary Data 2). The identity and function of the cDNA clones were predicted by searching the NCBI database and the pfam database. In total, we identified 142 MS2-interacting protein genes (Supplementary Data 10).

We tested the interaction between MS2 and eight representative full-length interactors, which included the eukaryotic elongation factors eEF1A, five ribosomal proteins (RPS2, RPS9, RPS25, RPL15, and RPL17), the GTP-binding protein (GTPase), and the ubiquitin-conjugating enzymes Ubc4B (Supplementary Table 3; Supplementary Data 10). We also tested the pairwise interaction between wild-type Ms2 and its four single-residue variants, including Ms2$^{L44F}$, Ms2$^{T109M}$, Ms2$^{A153V}$ and Ms2$^{E248K}$ (Supplementary Data 5), which confer male-sterility in EMS mutants. During pairwise interactions, wXb12 (PC332) and wXb12IP2 (PC324) were used as a positive control; wXb12 and wRAR1 (PC322) were used as a negative control (Supplementary Table 3)[59].

For MS2-interacting proteins, we performed GO analysis using SEA and the parametric analysis of gene set enrichment (PAGE)[58] and prepared a heat map (Supplementary Fig. 13d). Both a protein ID and its count in the cDNA library screening were submitted for the PAGE analysis.

**mRNA in situ hybridization.** Tissue preparation and *in situ* hybridization was performed according to a standard protocol[61]. LM15$_{RMs2}$ anthers at different stages were harvested and fixed in FAA (10% formaldehyde:5% acetic acid:50% alcohol) for 12 h at 4 °C. Probe templates were amplified from the *Ms2* cDNA using PCR primers P146 and P147. Both sense and antisense RNA probes were prepared independently using the digoxigenin RNA labelling kit (Roche Diagnostics, Mannheim, Germany).

**MS2 diversity in Triticeae species.** To assess genetic diversity of the *MS2* gene, we genotyped 575 germplasm, including 469 common wheat lines, 36 SHW lines and 70 *Ae. tauschii* accessions (Table 1; Supplementary Data 8). For the *MS2* gene, we developed four haplotype markers (HT1, HT2, HT3 and HT4; Supplementary Data 3) which target the polymorphic sites located at nucleotide positions $-673..-364$, $-314..-310$, 902..905 and 3404, respectively (Supplementary Data 3). The first three haplotype markers were amplified for 38 cycles with annealing temperatures of 67, 63, and 64 °C, respectively. HT4 was amplified for 35 cycles with an annealing temperature of 62 °C, and its PCR product was cleaved using the restriction enzyme *Alu*I (New England Biolabs). Their PCR products were separated on a 1.5–2% agarose gel.

**Data availability.** The DNA sequence data have been deposited in GenBank: KX533929 (*Ms2* cDNA), KX533930 (*ms2* cDNA), KX585234 (*Ms2*-associated BAC sequence) and KX585235 (*ms2*-associated BAC sequence). The RNA-seq data have been deposited in the NCBI Sequence Read Archive under accession number SRP092366. Other data supporting the findings of this study are either within the article and the Supplementary Information or available on request from the corresponding author. Plant transformation vectors are available with a material transfer agreement. For overseas requests, transgenic plants and Taigu lines will be provided if an export permit is issued by the Chinese Ministry of Agriculture on a case-by-case basis.

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

## Acknowledgements

This work was supported by the National Natural Science Foundation of China (31471161 and 31110103917) and the Natural Science Foundation of Shandong Province, China (JQ201107). We thank Drs Cathy M. Wilson and Allan Caplan for reviewing the manuscript and Dr Lei Ge for help with microscopy.

## Author contributions

D.F. conceived the project; M.-C.L., L.E., H.W., X. Zhang, C.C. and L.S. contributed ideas and resources, F.N., J.Q., Q.H., B.L., Y.W., S.W., F.C., C.Z. and X. Zhao performed the experiments; D.F. and F.N. analysed the data; D.F., F.N. and L.E. wrote the manuscript; and all authors discussed the results and the manuscript.

## Additional information

**Competing interests:** Shandong Agricultural University has filed patent applications (201510303817.0 and PCT/IB2016/000537) on behalf of F.N., J.Q., B.L., S.W. and D.F. regarding the *Ms2* gene and its promoter. The remaining authors declare no competing financial interests.

