## [Peer Review File · Nature Communications]

Reviewers' comments:

Reviewer #1 (Remarks to the Author):

The male sterile gene Ms2 in the Taigu line, which is a dominant locus, is widely used in wheat breeding for decades. This study reported the isolation and characterization of Ms2 by map-based cloning. They found that Ms2 is an orphan gene in the Triticinae. Ms2 is expressed specifically in anthers, and this expression is associated with an insertion of a novel retroelement into the promoter. Therefore, Ms2 is a gain-of-function new gene activated by this retroelement insertion. Transformation of Ms2 into wheat, barley and Brachypodium confirmed its dominant function on male sterility. This study also screened a number of Ms2-interactors, some of which are involved in translational machinery. This work is important and interesting, especially for new gene origination and neofunctionalization, but there remain a few questions or concerns. The manuscript is generally well written.

1. What is the expression pattern of PG5p1076F (or other fertile allele without the retroelement insertion), and does it have a primary function ?
2. Can the protein from PG5p1076F interact with those Ms2-interactors ?
3. The questions 1 and 2 can raise a new question that was Ms2 generated only by the retroelement insertion, or besides the retroelement insertion the nucleotide variations in the coding region also is important ?
4. What is the reason for activation of Ms2 by the retroelement insertion? Does the inserted retroelement contain a transcription enhancer, or this insertion destroyed a repressive cis-element in the promoter region?
5. The authors should observe by TUNEL assay if PCD process in the tapetum of different stage anthers of the Ms2 line and fertile line are different.
6. (minor points) the dominant feature of Ms2 should be indicated in abstract and introduction, not in the result (line 115). For example, in abstract, change "The male-sterile gene Ms2" to "The dominant male-sterile gene Ms2", and in line 43, revised as "Ms2 confers genetically dominant male-sterility in hexaploid and tetraploid wheat16".

Reviewer #2 (Remarks to the Author):

REVIEW

Wheat Ms2 encodes for a novel protein that confers male sterility in grass species

Overall a nice manuscript describing the convincing identification of the gene under Ms2 male sterility in wheat. The complementation is good and haplotype analysis supports the conclusion that the gene has been identified. The mutation analysis is very confusing and not well documented. A clear delineation of tracking mutations in M1 to M2 and M3 families is needed. The bulk analysis of M2 doesn't make much sense and adds confusion if the mutations are confirming the Ms2 phenotype or not.

The global expression analysis is interesting but not much description of what are the implications of apparent total remodeling of the exome in anthers due to Ms2.

The described use in hybrid breeding and hybrid seed productions is rudimentary and lacking. The formation of single plant hybrids doesn't make much sense and would not be much use to breeders to evaluate single plants for which the heritability is too low. Need a more useful description of the potential use in breeding and producing hybrid wheat (there is huge potential! But not as described here. Recommend discussing with some breeders.)

The end of the manuscript is not strong. Needs a concluding paragraph to put all findings into larger context.

L34 – probably need to add a short comment backing the statement that CHA hybrids are inferior.

L55 – 'molecular farming' is a strange phrase. Should remove.

L70 – "The stages of anther development in wild-type wheat and Taigu lines are shown in the Fig. 1." This is not a useful sentence. Remove.

L73 – need to clarify what 5.1% and 14.2% SNPs means. This is the per base polymorphisms rate??? Need to clarify that this is from the 90K array?

L97 – I don't think you can say only candidate, but 'strongest' candidate.

L118-133 – The mutation work doesn't make much sense as described. Why were the M2 plants with an M1 lineage not evaluated for co-segregation of mutations in PG5 mutations?

L185 – what do these numbers mean? Could use a better designation to clarify SNPs, indels, etc.

L197 – "appeared to be the most variable, representing an evolutionary hotspot." Need something more substantial that appeared' to make this statement and should describe what is considered an 'evolutionary hotspot' if saying this.

Supplemental Figures need 'S#' designations.

Figure S14 – fix spelling of 'production'

L203-205 – The functionality of this recombinant gene warrants some discussion.

Single plant screening for high-yielding hybrids is not a useful approach. Perhaps making the hybrid seed and then evaluating for yield in replicated plots, but that is different that what is proposed here.

L243 – hybrid seed production would not be for the most desirable heterosis. The heterosis and hybrid performance per se, would have been evaluated in the breeding of the hybrid parents. An A / B / R type approach could be used here to produce the hybrid seed, but that is not described.

Figure 3 – correct 'Golden Promise'

It is hard to believe that the Ms2 expression (Figure 4) is that tightly regulated to anther specific expression and also to just S2 stage in the anthers. Not any detectable expression in any other tissues???

Reviewer #3 (Remarks to the Author):

This manuscript describes the identification of the wheat Male Sterile 2 gene, currently used in hybrid wheat production, and the demonstration that this gene can be used to induce male sterility in wheat and other grasses. Based on their results the authors suggest a GM (potentially cisgenic) scheme to engineer male sterility for use in identifying superior heterosis and producing hybrid seeds for commercial use.

The manuscript represents an impressive amount of work that clearly demonstrates that the correct Ms2 gene has been identified. Subsequent experiments suggests that the encoded protein may be acting to affect translation or related essential functions to block normal anther development and pollen production.

Although the manuscript is well written, it is quite complex and I suspect hard to follow for those not already familiar with the wheat hybrid systems. For this reason I suggest that the introduction be expanded to include an explanation of how Ms2 is currently used for hybrid seed production - this would also make it easier to follow the GM approach described in the Discussion.

Additional comment: On line 165 the authors state that the 'terminator region is not involved in regulating Ms2 expression'. This does not logically follow from the data presented, and should be rewritten as 'terminator region is not required for induction of male sterility, suggesting that it does not have a critical role in regulating Ms2 expression'.

Point-by-point responses to the reviewers' comments of manuscript NCOMMS-16-20359-T

Reviewer #1

The male sterile gene Ms2 in the Taigu line, which is a dominant locus, is widely used in wheat breeding for decades. This study reported the isolation and characterization of Ms2 by map-based cloning. They found that Ms2 is an orphan gene in the Triticinae. Ms2 is expressed specifically in anthers, and this expression is associated with an insertion of a novel retroelement into the promoter. Therefore, Ms2 is a gain-of-function new gene activated by this retroelement insertion. Transformation of Ms2 into wheat, barley and Brachypodium confirmed its dominant function on male sterility. This study also screened a number of Ms2-interactors, some of which are involved in translational machinery. This work is important and interesting, especially for new gene origination and neofunctionalization, but there remain a few questions or concerns. The manuscript is generally well written.

1. What is the expression pattern of PG5_{P1076F} (or other fertile allele without the retroelement insertion), and does it has a primary function?

Response: We have never detected transcription of *PG5_{P1076F}* and other fertile alleles in wild type male-fertile wheat, suggesting that fertile alleles are silent. Specifically, we have tested LM15 for *PG5* using 36-cycle RT-PCR with conserved primers for both *PG5_{P1593S}* and *PG5_{P1076F}*; no band was detected. We have done this experiment in approximately five trials that were conducted at separate times, each with independently collected tissue. Tissues examined include anther, pistil, leaf, shoot, root, and so on.

To show this, we added the negative control (LM15) to Figure 4a and 4b. Similarly, for the *in situ* hybridization, we replaced the Figure 4c-c4 with a negative control.

We also reported in the paper that “*PG5_{P1593S}* encodes a 293-aa protein (31.5 kDa, pI 5.7) with no annotated domains or protein homologs in GenBank.” We used *PG5* cDNAs to search expressed sequence tags in GenBank, but there was no close homologue, suggesting that the transcription of *PG5* has not been detected worldwide.

In addition, we deposited our RNA-seq data in the NCBI Sequence Read Archive (accession number SRP092366). In total, there are 272 cDNA reads of *PG5* in the male-sterile anther in LM15_{RM_s2}, but there were no cDNA reads of *PG5* in male-fertile anthers of LM15 and in flag leaves of LM15_{RM_s2} and LM15. Consequently, *PG5_{P1076F}*

remains silent in the wild type male-fertile wheat.

Therefore, we think that these silent alleles in male-fertile wheat have no primary function. There are also seven *Aegilops tauschii* accessions without the *MS2* (or *PG5*) gene (Table 1), suggesting that *PG5_{P1076F}* and other fertile alleles have no primary function in plants.

Added text in blue: “In male-fertile wheat, *MS2* (e.g. *PG5_{P1076F}*) mRNA was not expressed in all vegetative and reproductive tissues tested (Fig. 4a), suggesting that the male-fertile *MS2* has no primary function.”

Revised legend in blue in Figure 4: “(a,b) *Ms2* displays anther-specific transcription in the S2 stage. *BW_{Ms2}*, *BW_{Ms2:GFP}*, and *GP_{Ms2}* are male-sterile transgenic plants JZ7-2, JZ19-6, and 22-1A, respectively. GLP was pooled from glumes, lemma, and palea. A 36-cycle PCR was performed to detect transcripts of *Ms2* (P129/P130, 997 bp) and *ACTIN* (P142/P143, 503 bp). (c) *In situ* hybridization of *Ms2* mRNA in cross sections of LM15_{*RM_{s2}*} stamens. Hybridization with antisense probes was detected in pollen mother cells (PMC, c1), early uninucleate microspores (c2), and late uninucleate microspores (c3); hybridization with sense probes was negative in uninucleate microspores (c4).”

Revised Figure 4: we added negative control (LM15) in a and b; we replaced c4 using a negative control.

2. Can the protein from PG5_{P1076F} interact with those Ms2-interactors ?

Response: Yes. We identified 142 proteins, of which 64 proteins only interacted with *Ms2* (=PG5_{P1593S}), four proteins only interacted with *ms2* (=PG5_{P1076F}), and 74 proteins interacted with both *Ms2* (=PG5_{P1593S}) and *ms2* (=PG5_{P1076F}). This data is in Figure 5c and the Supplementary Table 13.

3. The questions 1 and 2 can raise a new question that was Ms2 generated only by the retroelement insertion, or besides the retroelement insertion the nucleotide variations in the coding region also is important ?

Response: Based on our data, we think that the dominant *Ms2* was generated by the retroelement insertion. We don't see any compelling evidence that the nucleotide variation in the coding region contributed to the formation of the dominant *Ms2*. There are three lines of evidence supporting this conclusion.

First, the retroelement is the **only** difference between a dominant *Ms2* (*PG5_{P1593S}*) in Taigu lines and a recessive *ms2* in ‘Chinese Spring’ (Supplementary Table 4). This information is now stated: “In the CS haplogroup, *Ms2* has a Taigu retroelement

(inserted at nucleotide position -314..-307), but the male-fertile *ms2* does not: the only difference between these two alleles is the *Taigu* retroelement.”

Second, we now stated in the manuscript: “Despite some nucleotide differences between *PG5_{P1593S}* and *PG5_{P1076F}*, *PG5_{P1076F}* appears to be functionally equivalent to *PG5_{P1593S}*, because the overexpression of either *PG5_{P1593S}* or *PG5_{P1076F}* apparently prevents generation of transgenic plants. We postulate that the key difference between the MS and MF alleles is that the MS allele is expressed and the MF allele is not.”

Third, the newly formed *Ms2_{O261S}* inherited its promoter and first two exons from the dominant *Ms2* in XY6_{*Ms2*}, but the rest of the gene comes from a recessive *ms2* in SW41 (Supplementary Table 4). We then stated in the manuscript: “Despite having a large portion of the recessive *ms2* allele and the T3404A mutation, the hybrid *Ms2_{O261S}* gene confers male-sterility, which demonstrates that the nucleotide differences between MS and MF alleles in the third to last exons have no effect on *PG5* function.”

Therefore, “We conclude that *Ms2* (Haplotype A2) was formed when the retrotransposon *Taigu* inserted into the A1 haplotype.”

4. What is the reason for activation of *Ms2* by the retroelement insertion? Does the inserted retroelement contain a transcription enhancer, or this insertion destroyed a repressive *cis*-element in the promoter region?

Response: This is a great question. We have been thinking about how a silent gene became active, either due to *Taigu* (a LTR-type retrotransposon)-associated transcription enhancer or due to the disruption of a repressive *cis*-element.

As we addressed in the paper (and now edited in blue), “Of the four regions investigated, we compared the number of accessions in three genotype groups (A, B, and a ‘non-AB’ group, Table 1) of all materials except for those in the null group (n=546). A Chi-square test revealed no significant difference amongst HT1, HT3, and HT4, but HT2 was associated with a higher number of ‘non-AB’ accessions than those in either HT1, HT3, or HT4 ($\chi^2 > 35.2$, $df=2$, $P < 2.2 \times 10^{-8}$). Therefore, the polymorphic site -314..-307 (HT2) is the most variable and might be a region that is more likely to be involved in genetic change.”

We then further discussed the potential role of the *Taigu* retroelement by adding the following section in the supplementary discussion.

2.3 The *Taigu* retroelement may act as a transcription enhancer

Plant transposable elements (TEs) can have multiple elements that regulate gene expression²³. In LTR-type retrotransposons, the 5’ LTR usually contains a promoter for producing template RNAs²¹. The retrotransposon/LTR-based promoter can

directly induce gene expression²² or can play a role in activating ‘sleeping’ genes²⁴. LTR retrotransposons also can have finely tuned responses to a diverse array of external stimuli, and can act as dispersed regulatory modules that can respond to external stimuli and activate adjacent plant genes²⁵.

In this study, we do not know whether the *Taigu* retroelement acts as transcription enhancer or its insertion destroys a repressive *cis*-element. If there were a repressive *cis*-element in the promoter region of the *MS2* gene, other sequence variations of the *cis*-element would likely also activate the *MS2* gene. If so, we would expect to see more dominant male-sterility genes mapped to the wheat 4DS chromosome. However, this is not the case. For example, the H2 and H3 haplotypes (e.g. ‘Clae 30’ and ‘AL8/78’, Supplementary Table 11) are associated with a 49-bp deletion in close vicinity (4bp) to the insertion site of the *Taigu* retroelement, but accessions of the H2 and H3 haplotypes are male fertile. Consequently, we hypothesize that the *Taigu* retroelement acts as a transcription enhancer and activates the transcription of the *MS2* gene.

Revised text in blue: “We postulated that the *Taigu* retroelement acts as a transcription enhancer that activates transcription of *PG5_{P1593S}* rather than an inactivator of a repressive *cis*-element in the promoter region (SI 2.3).”

5. The authors should observe by TUNEL assay if PCD process in the tapetum of different stage anthers of the *Ms2* line and fertile line are different.

Response: We appreciate this constructive suggestion to compare the PCD process in the tapetum in male-sterile and male-fertile lines from *Taigu* wheat progeny. We would like to do the experiment! However, here we first argue that the PCD results will neither affect the key results of the current MS, nor answer many of the remaining mechanistic questions. We note that the word “tapetum” only appears in the main text twice: once to state that it was one of the anther cell types in which we detected *Ms2* mRNA; and secondly to cite other work that demonstrates that the tapetum and other anther cells eventually degenerate via PCD. Therefore, we are not intending to draw any direct link between tapetum PCD and male sterility in *Taigu* lines, and we did not intend to identify tapetum as the key cell that results in the obvious stoppage of anther development in *Taigu* lines. In *Taigu* lines, we observed premature declines in multiple cell types: 1) the middle layer starts to degenerate as early as the sporogenous cell stage, and 2) any uninucleate microspores that are produced collapse -- the collapse is correlated with the tissue-specific expression of *Ms2* in the middle layer and the male reproductive cells. Obviously, anther degeneration in *Taigu* lines is complex, which requires a comprehensive study to understand the underlying mechanisms.

While we believe that the TUNEL assay would be an excellent approach to assess the PCD process in anther development, there are three practical problems that prevent us

from accomplishing the TUNEL assay during the limited time period for resubmission. First, I as the corresponding author am currently relocating from China to the University of Idaho in United States, and I am still at an early point in establishing a new lab with new personnel. Secondly, a successful TUNEL assay requires excellent skills for preparation of semi-thin sections of anthers. In this study, the co-author Yan Wang spent nearly three years studying the microsporogenesis process of Taigu lines (Supplementary Figure 1). However, Yan Wang has graduated from my previous lab, and there is no other personnel qualified to do this job. Third, winter wheat grows slowly. If we planted wheat seeds today, it would take at least four months to be ready for the first sampling. Sampling, section preparation, the TUNEL assay, and an experimental repeat would take at least two more months. Therefore, if everything goes well, it would take at least half a year to accomplish this experiment. Consequently, it is not possible to accomplish the experiment within three months that is asked by the journal. Since 2010, we have been focusing on the cloning of the *Ms2* gene, and were lucky to get the cloning job done in seven years. However, research in this field is extremely competitive, and we do not want to postpone publication for a minimum of another six months.

Finally on this point, our primary objectives were to clone the *Ms2* gene and to determine its gene diversity in wheat and its D-genome progenitors. First, map-based positional cloning has been very important for identifying agronomically important genes in wheat. For example, *Nature Genetics* recently reported the map-based cloning of the resistance gene *Fhb1* for wheat *Fusarium* head blight (doi:10.1038/ng.3706). In the current study, we demonstrated the cloning of *Ms2* using high-density mapping, genetic complementation, and mutagenesis. Second, we performed a comprehensive analysis on haplotypes of the *MS2* region and the *MS2* gene. It is now clear that *MS2* originated in the Triticeae and became active after a long-term of evolution.

To conclude this point, while we did not finish the research on the *Ms2* gene, we believe that we have made a significant contribution, and consequently, we hope that the reviewers will agree that the request for a TUNEL assay would have been good, but is not necessary for publication.

6. (minor points) the dominant feature of *Ms2* should be indicated in abstract and introduction, not in the result (line 115). For example, in abstract, change “The male-sterile gene *Ms2*” to “The dominant male-sterile gene *Ms2*”, and in line 43, revised as “*Ms2* confers genetically dominant male-sterility in hexaploid and tetraploid wheat16”.

Response: We agree.

Revised text: We changed “The male-sterile gene *Ms2*” to “The dominant male-sterile gene *Ms2*” in the abstract. Furthermore, we changed “*Ms2* confers

male-sterility in hexaploid and tetraploid wheat” to “*Ms2* confers genetically dominant male-sterility in hexaploid and tetraploid wheat” in the manuscript text.

Reviewer #2 (Remarks to the Author):

*Wheat *Ms2* encodes for a novel protein that confers male sterility in grass species. Overall a nice manuscript describing the convincing identification of the gene under *Ms2* male sterility in wheat. The complementation is good and haplotype analysis support the conclusion that the gene has been identified.*

*1. The mutation analysis is very confusing and not well documents. A clear delineation of tracking mutations in M1 to M2 and M3 families is needed. The bulk analysis of M2 doesn't make much sense and adds confusion if the mutations are confirming the *Ms2* phenotype or not.*

Response: We apologize for writing unclearly, and have re-written this section to address both Reviewer no. 1 and 2's comments. Early on, male-fertile mutants were clustered based on specific base changes; some clusters had multiple mutant spikes/lines with the same mutation. We have revised Supplementary Table 5 to better show this. No M₃ were analyzed. We'd like to make two points. 1) While the mutational analysis was strongly supportive of the contention that *PG5_{P1593S}* is *Ms2*, we used the cloning of *PG5_{P1593S}* and transformation of three crops to prove that *PG5_{P1593S}* is *Ms2*, and not the mutational analysis. 2) As now indicated hopefully better in the text, the genetics of male sterility (*Ms2ms2*) is such that all progeny of a male-fertile M₁ produce male fertile M₂, regardless of whether they have a mutated *PG5_{P1593S}* or two copies of *ms2*. However, the mutated *PG5_{P1593S}* allele was associated with loss of male sterility in M₁ to M₂ to M₃ except when the mutated *PG5_{P1593S}* allele was backcrossed to a Taigu line with a wild-type *PG5_{P1593S}*.

Revised text in blue: “*Ms2* is dominant for male-sterility (MS), in keeping with the dominant expression of *PG5_{P1593S}* in Taigu lines; all Taigu lines are heterozygous for *Ms2* because two Taigu lines cannot cross. We chemically mutagenized LM15_{RM_{s2}} and CS_{RM_{s2}} to determine if any EMS mutants had 1) a male-fertile (MF) phenotype and if so, 2) a variation in their *PG5_{P1593S}* (SI 3.3) that would explain the transformation from MS to MF. We first generated 2,266 M₁ plants and examined 37,471 spikes, of which 254 had MF anthers and viable pollen (Fig. 3a,b). Amongst all the EMS-treated LM15_{RM_{s2}} M₁ sampled (n=1,200), 2.9% of the primary tillers had a TILLING mutation in *PG5_{P1593S}*, compared to a 40% mutation rate in *PG5_{P1593S}* in the induced MF tillers (n=20). A Chi-square test negated the independence between *PG5_{P1593S}* mutations and male fertility in LM15_{RM_{s2}} ($\chi^2=97.1$, df=1, $P=6.5 \times 10^{-23}$), i.e., EMS-induced mutation(s) in *PG5_{P1593S}* are associated with the loss of male sterility. Amongst all the EMS-treated CS_{RM_{s2}} M₁ sampled (n=1,066), we identified 234 MF

spikes in 149 M₁ plants (Supplementary Fig. 8). We selected 178 MF M₁ tillers and examined the length of PG5_{P1593S}; 29% had either a partial or complete deletion in PG5_{P1593S}. We then selected 111 of the MF M₁ tillers and sequenced the first two exons of PG5_{P1593S} (Supplementary Table 5); 23% had point mutations that caused either a residue change or a truncation in the PG5_{P1593S} protein. We then selected 99 MF M₁ to represent the range of mutations, selfed them, and examined the M₂ (Fig. 3a,b); all were male-fertile, and all had inherited the mutation from the M₁. We also identified M₂ mutants that were associated with insertion/deletion in the PG5_{P1593S} protein (Supplementary Table 5). Thus, PG5_{P1593S} likely confers male-sterility in Taigu lines.”

2. The global expression analysis is interesting but not much description of what are the implications of apparent total remodeling of the exome in anthers due to Ms2.

Response: This is a good point. We had some information in Supplementary Table 12, We have added a new section in the supplementary results.

Added section in the supplementary results:

1.6 MS2-based transcriptome in MS and MF anthers

Using RNA-seq analysis, we identified 2,942 sterile-anther-enriched genes (SAEGs) and 1,011 fertile-anther-enriched genes (FAEGs) (Fig. 5b). Among them, 1,847 SAEGs were in the GO database and revealed 119 significant GO terms, and 537 FAEGs revealed 58 significant GO terms (Supplementary Table 12).

In the FAEG group, many of the genes are involved in metabolism (GO:0008152, $P = 1 \times 10^{-15}$), photosynthesis (GO:0015979, $P = 2 \times 10^{-77}$), translation (GO:0006412, $P = 2 \times 10^{-9}$), and protein folding (GO:0006457, $P = 6 \times 10^{-5}$). Based on molecular function the structural constituent of ribosome had the strongest significance in the FAEG group (GO:0003735, $P = 6 \times 10^{-15}$). Remarkably, the FAEG group also hit GO terms for cellular components, including the macromolecular complex (GO:0032991, $P = 2 \times 10^{-11}$), membrane (GO:0016020, $P = 6 \times 10^{-9}$), thylakoid (GO:0009579, $P = 3 \times 10^{-26}$), ribosome (GO:0005840, $P = 6 \times 10^{-15}$) and intracellular non-membrane-bounded organelle (GO:0043232, $P = 1 \times 10^{-7}$). Taken together, GO analysis supported active photosynthesis in the MF anthers, which are green, which likely drives the translation process and contributes to the synthesis of ribosomes, translation factors, and other components of the translation machinery.

In the SAEG group, there were also many genes involved in metabolism (GO:0008152, $P = 6 \times 10^{-17}$), oxidation reduction (GO:0055114, $P = 3 \times 10^{-13}$), polysaccharide catabolism (GO:0000272, $P = 1 \times 10^{-9}$), and protein modification

(GO:0006464, \$P = 5 \times 10^{-12}\$ ). Other SAEG genes may impact cell wall organization (GO:0071554, \$P = 4 \times 10^{-11}\$ ) and encode proteins that can digest cell wall macromolecules (GO:0016998, \$P = 5 \times 10^{-16}\$ ). As for their molecular function, most SAEG genes were associated with specific catalytic activities (GO:0003824, \$P = 6 \times 10^{-26}\$ ), such as oxidoreductase (GO:0016491, \$P = 9 \times 10^{-14}\$ ), transferase (GO:0016740, \$P = 4 \times 10^{-12}\$ ), protein kinase (GO:0004672, \$P = 5 \times 10^{-17}\$ ), and chitinase (GO:0004568, \$P = 4 \times 10^{-13}\$ ). Some SAEG products may bind to ion (GO:0043167, \$P = 6 \times 10^{-8}\$ ), tetrapyrrole (GO:0046906, \$P = 2 \times 10^{-26}\$ ), heme (GO:0020037, \$P = 2 \times 10^{-29}\$ ), polysaccharide (GO:0030247, \$P = 7 \times 10^{-11}\$ ), and sequence-specific DNA (GO:0043565, \$P = 3 \times 10^{-6}\$ ). However, there was no significant hit on the cellular components for all SAEG genes. We postulate that the Ms2 protein causes a cellular chaos that impacts cell fate. In comparison, the MF anthers are active in energy production, protein translation, protein folding, but these cellular processes are inhibited in the MS anthers. Most likely, Ms2 acts as a direct or indirect factor that impacts photosynthesis, translation, and/or other cellular activities, which in turn causes pre-mature degeneration in the MS anthers.

3. The described use in hybrid breeding and hybrid seed productions is rudimentary and lacking. The formation of single plant hybrids doesn't make much sense and would not be much use to breeders to evaluate single plants for which the heritability is too low. Need a more useful description of the potential use in breeding and producing hybrid wheat (there is huge potential! But not as described here. Recommend discussing with some breeders.)

Response: Thanks for the comment. We agree that the current description on hybrid breeding (the original Fig. 14) lacks experimental evidence. For this reason, we introduced our idea by saying “Here, we propose a high-throughput hybrid production system ...”

Based on the reviewer's suggestion, we discussed with wheat breeders about how to use the Ms2 gene in breeding. Interesting, many breeders think the Ms2 gene will play an important role in wheat recurrent selection for new varieties. We hope the new paragraph is better description of the potential use of the Ms2 gene in wheat breeding.

Added paragraph in the discussion section:

“We have cloned the first male-sterile gene in wheat. In China, the Ms2 gene has been widely used during recurrent selection in conventional breeding of wheat²⁴. The cloning of the dominant Ms2 will boost recurrent selection in wheat globally and will enable additional applications for population improvement and gene pyramiding in autogamous crops²¹. For instance, any transformation-amenable wheat can be engineered to carry the Ms2 gene, which will facilitate a Ms2-based recurrent selection in the ‘engineered Taigu’ (eTaigu) line (Supplementary Fig. 14). Although eTaigu lines would be classified as either cisgenic or transgenic, the final cultivars

would have male-fertility restored and be *Ms2*-free. Consequently, use of the *Ms2* gene may be more readily accepted by the public. Here, we also demonstrated that *Ms2* confers male-sterility in barley and *Brachypodium*, in addition to wheat. The range of crops in which the transgenic *Ms2* could function is unknown. Nonetheless, the potential for cereal and perhaps other crop improvement is great.”

Added legend for a new figure:

Figure 14 | *Ms2*-based recurrent selection in wheat. This procedure, adapted from previous studies^{3,39-41}, includes the use of either wild-type Taigu or ‘engineered Taigu’ (eTaigu) lines that have a *Ms2* gene. The procedure includes the development of an initial “base” population (green), the development and maintenance of recurrent populations (gray), and the release of advanced lines and cultivars (light blue). The number of steps in the simplest path is indicated in the center. The number of steps in more complicated paths is marked by a ‘greater than or equal to’ sign (\geq). A solid step line indicates bulk seeds from selected male sterile (MS) plants and a dashed step line indicates seeds from selected male fertile (MF) plants. Seeds of MF plants can be used to maintain the recurrent populations (dashed lines in gray) and to develop advanced lines and cultivars (dashed lines in light blue). To enrich the genetic variability in recurrent populations, novel genotypes/lines can be added into the recurrent populations, as marked by a plus sign (+). During the course of recurrent selection and progeny tests, undesirable plants/lines will be removed, as labeled by an x-shaped cross.

4. The end of the manuscript is not strong. Needs a concluding paragraph to put all findings into larger context.

Response: Thanks! This is an excellent comment. We added a concluding paragraph.

Added paragraph in the discussion section:

“In conclusion, we report the first molecular characterization of a male-sterility gene in wheat. The current *Ms2* gene has played pivotal roles in wheat breeding, promoting the release of hundreds of breeding lines/cultivars in common wheat. Remarkably, *Ms2* homologs are only found in wheat and its close relatives, and are absent from other crops outside the Triticeae tribe. The *Ms2* protein likely represses plant translation machinery, documenting a novel strategy for generating male sterility. A cloned *Ms2* gene provides high value for breeding and producing hybrid wheat, and potentially for other major crops.”

5. L34 – probably need to add a short comment backing the statement that CHA

hybrids are inferior.

Response: We added a short comment.

Added text in blue: “which adds cost and furthermore, results in inferior hybrids with either poor germination or reduced seedling vigor¹¹.”

6. L55 – ‘molecular farming’ is a strange phrase. Should remove.

Response: We deleted “molecular farming” and revised the original sentence.

7. L70 – “The stages of anther development in wild-type wheat and Taigu lines are shown in the Fig. 1.” This is not a useful sentence. Remove.

Response: We have deleted this sentence.

8. L73 – need to clarify what 5.1% and 14.2% SNPs means. This is the per base polymorphisms rate??? Need to clarify that this is from the 90K array?

Response: We are sorry for the confusion. This is not the per base polymorphisms rate, but the polymorphic SNPs rate on the wheat 90K iSelect SNP assay. We revised the original sentence.

Revised text in blue: We used synthetic hexaploid wheat (Supplementary Table 1) to enrich for D genome SNPs. Based on the wheat 90k SNP chip, Taigu lines had 5.1% and 14.2% SNPs with SW7 and SW41, respectively (Supplementary Fig. 3), which was sufficient for mapping and cloning *Ms2*.

9. L97 – I don’t think you can say only candidate, but ‘strongest’ candidate.

Response: We replaced ‘only’ by ‘strongest’.

10. L118-133 – The mutation work doesn’t make much sense as described. Why were the M2 plants with an M1 lineage not evaluated for co-segregation of mutations in PG5 mutations?

Response: We apologize for the insufficient description. In Taigu lines, a dominant *Ms2* gene for male sterility always remains heterozygous; its recessive homolog *ms2*

acts as a null allele. When a disrupting mutation occurred to the dominant *Ms2* in a M_1 lineage, only the mutated *Ms2* and the recessive *ms2* were passed to M_2 during self-pollination, and all M_2 plants were male-fertile. Consequently, we did not evaluate the co-segregation of *PG5* mutations among M_2 plants of the same M_1 lineage. To make it clear, we revised the manuscript and the Supplementary Table 5.

Revised text in blue: “*Ms2* is dominant for male-sterility (MS), in keeping with the dominant expression of *PG5*_{*P1593S*} in Taigu lines; all Taigu lines are heterozygous for *Ms2* because two Taigu lines cannot cross. We chemically mutagenized LM15_{*RM_{s2}*} and CS_{*RM_{s2}*} to determine if any EMS mutants had 1) a male-fertile (MF) phenotype and if so, 2) a variation in their *PG5*_{*P1593S*} (SI 3.3) that would explain the transformation from MS to MF. We first generated 2,266 M_1 plants and examined 37,471 spikes, of which 254 had MF anthers and viable pollen (Fig. 3a,b). Amongst all the EMS-treated LM15_{*RM_{s2}*} M_1 sampled (n=1,200), 2.9% of the primary tillers had a TILLING mutation in *PG5*_{*P1593S*}, compared to a 40% mutation rate in *PG5*_{*P1593S*} in the induced MF tillers (n=20). A Chi-square test negated the independence between *PG5*_{*P1593S*} mutations and male fertility in LM15_{*RM_{s2}*} ($\chi^2=97.1$, $df=1$, $P=6.5 \times 10^{-23}$). i.e., EMS-induced mutation(s) in *PG5*_{*P1593S*} are associated with the loss of male sterility. Amongst all the EMS-treated CS_{*RM_{s2}*} M_1 sampled (n=1,066), we identified 234 MF spikes in 149 M_1 plants (Supplementary Fig. 8). We selected 178 MF M_1 tillers and examined the length of *PG5*_{*P1593S*}; 29% had either partial or complete deletion in *PG5*_{*P1593S*}. We then selected 111 of the MF M_1 tillers and sequenced the first two exons of *PG5*_{*P1593S*} (Supplementary Table 5); 23% had point mutations that caused either a residue change or a truncation in the *PG5*_{*P1593S*} protein. We then selected 99 MF M_1 to represent the range of mutations, selfed them, and examined the M_2 (Fig. 3a,b); all were male-fertile, and all had inherited the mutation from the M_1 . We also identified M_2 mutants that were associated with insertion/deletion in the *PG5*_{*P1593S*} protein (Supplementary Table 5). Thus, *PG5*_{*P1593S*} likely confers male-sterility in Taigu lines.”

11. L185 – what do these numbers mean? Could use a better designation to clarify SNPs, indels, etc.

Response: They are nucleotide position on the genomic *PG5* gene in the common wheat ‘Chinese Spring’ (Supplementary Table 4). We added ‘nucleotide position’ to those numbers. For this reason, we did not try to find a better designation to clarify different polymorphic sites such as SNPs and InDels.

Revised text in blue: “In the CS haplogroup, *Ms2* has a Taigu retroelement (inserted at nucleotide position -314..-307), but the male-fertile *ms2* does not.”

“one in the promoter (at nucleotide position -673..-364) and one in the second intron (at nucleotide position 902..905). We further genotyped 575 assessments using four haplotype markers, which target polymorphic sites located at nucleotide positions

-673..-364, -314..-307, 902..905 and 3404 (Table 1, Supplementary Tables 4, 10 and 11).”

“which target the polymorphic sites located at nucleotide positions - 673..- 364, - 314..- 307, 902..905 and 3404, respectively (Supplementary Table 4).”

Added legend text for the Table 1: “Here, we used two periods (..) to separate the starting and ending nucleotide position (Supplementary Table 4).”

Added legend text for the Supplementary Table 4: “Here, we used two periods (..) to separate the starting and ending nucleotide position.”

*12. L197 – “appeared to be the most variable, representing an evolutionary hotspot.”
Need something more substantial that appeared’ to make this statement and should describe what is considered an ‘evolutionary hotspot’ if saying this.*

Response: We revised the original sentence.

Revised text in blue: “Of the four regions investigated, the polymorphic site -314..-307 (HT2) is the most variable and might be a region that is more likely to be involved in genetic change.”

13. Supplemental Figures need ‘S#’ designations.

Response: We agree with the reviewer’s suggestion, but did not add ‘S#,’ because we believe that it does not fit the journal’s format.

14. Figure S14 – fix spelling of ‘production’

Response: Thanks, fixed.

15. L203-205 – The functionality of this recombinant gene warrants some discussion.

Response: Thanks for the comment. We added the following sentence.

Added text: “Therefore, the evolution of a molecular switch that turns on MS2 is essential for the functionalization of this gene.”

16. Single plant screening for high-yielding hybrids is not a useful approach. Perhaps making the hybrid seed and then evaluating for yield in replicated plots, but that is different than what is proposed here.

Response: Thanks for this comment. We revised the Supplementary Figure 15 and the description in manuscript text and figure legend.

Revised manuscript text in blue: “Here, we propose a high-throughput hybrid production system using any wheat that can be transformed with *Ms2* plus an aleurone-specific gene for pigmented kernels (MSC for *Ms2* color wheat) (Supplementary Fig. 15). The MSC system has two potential applications: 1) screening for desirable heterosis groups, and 2) hybrid seed production for either specific heterosis groups or, for example, for plants with new combinations of disease resistance genes.”

Revised legend in blue for the Supplementary Figure 15: “Wheat ‘A’ is first transformed to MSC wheat, amplified, and then planted in alternate rows. Within individual plots, pollen donors of a specific cultivar are planted in each row adjacent to the MSC wheat. All seed in the MSC rows will be a F_1 hybrid, and will be planted in the field and examined for desirable traits. F_1 s of each specific cross between one cultivar and an MSC line will be evaluated in replicated plots to identify specific heterotic groups of interest (dashed pink lines).”

17. L243 – hybrid seed production would not be for the most desirable heterosis. The heterosis and hybrid performance per se, would have been evaluated in the breeding of the hybrid parents. An A / B / R type approach could be used here to produce the hybrid seed, but that is not described.

Response: Yes. We agree that the heterosis and hybrid performance per se should be used in the breeding of the hybrid parents. We have revised the Supplementary Figure 15 and the description in manuscript text and figure legend.

Revised manuscript text in blue: “The MSC system has two potential applications: 1) screening for desirable heterosis groups, and 2) hybrid seed production for specific heterosis groups.”

Revised legend in blue for the Supplementary Figure 15: “The most promising heterotic groups from specific cultivars will be scaled up for hybrid seed production.”

18. Figure 3 – correct ‘Golden Promise’

Response: Revised. Thanks!

19. It is hard to believe that the Ms2 expression (Figure 4) is that tightly regulated to anther specific expression and also to just S2 stage in the anthers. Not any detectable expression in any other tissues???

Response: We appreciate the reviewer's concern, and indeed did not anticipate this result. However, we have repeated all of these results, and have looked for minimal levels of expression very carefully. Consequently, we are confident that the transcription of *MS2* is limited to a narrow developmental stage (e.g. S2) and specific in the anther tissue alone.

Related information is indicated in our comments to reviewer #1 point 4.

Reviewer #3 (Remarks to the Author):

This manuscript describes the identification of the wheat Male Sterile 2 gene, currently used in hybrid wheat production, and the demonstration that this gene can be used to induce male sterility in wheat and other grasses. Based on their results the authors suggest a GM (potentially cisgenic) scheme to engineer male sterility for use in identifying superior heterosis and producing hybrid seeds for commercial use.

The manuscript represents an impressive amount of work that clearly demonstrates that the correct Ms2 gene has been identified. Subsequent experiments suggests that the encoded protein may be acting to affect translation or related essential functions to block normal anther development and pollen production.

1. Although the manuscript is well written, it is quite complex and I suspect hard to follow for those not already familiar with the wheat hybrid systems. For this reason I suggest that the introduction be expanded to include an explanation of how Ms2 is currently used for hybrid seed production - this would also make it easier to follow the GM approach described in the Discussion.

Response: Thanks for this comment.

Added text in blue in the introduction section:

“If a male sterility gene were cloned, a genetic modification could be used to develop improved methods for improved hybrid production in major crops⁸.”

“Today both *Ms2*- and *RMs2*-based recurrent selection systems are widely used in

wheat-breeding programs in China during the recurrent selection phase. New wheat varieties are often bred in a ten or more generation process that includes recombination of parental lines (which is greatly assisted by male-sterility in an autogamous crop), selection of recombinants during the “recurrent selection” phase, followed by five or more generations of selfing.”

2. Additional comment: On line 165 the authors state that the 'terminator region is not involved in regulating Ms2 expression'. This does not logically follow from the data presented, and should be rewritten as 'terminator region is not required for induction of male sterility, suggesting that it does not have a critical role in regulating Ms2 expression'.

Response: Thanks! We have revised this sentence as suggested by the reviewer.

Reviewers' comments:

Reviewer #1 (Remarks to the Author):

This manuscript was revised properly with consideration of the comments.
minor comments:

Lines 81: Change 'S1 to S4' to S1 to S4 stages'

Line 1335: change "non-pollen" to "pollen-less"

Reviewer #3 (Remarks to the Author):

The authors have done a good job in responding to the reviewer's comments.

Reviewer #1

This manuscript was revised properly with consideration of the comments.

minor comments:

Lines 81: Change 'S1 to S4' to S1 to S4 stages'

Line 1335: change "non-pollen" to "pollen-less"

Response: Corrected. Thanks!

Reviewer #3

The authors have done a good job in responding to the reviewer's comments.

Response: Thank you!